# Inhibition of mutant RAS-RAF interaction by mimicking structural and dynamic properties of phosphorylated RAS

Metehan Ilter[1†‡], Ramazan Kasmer[2,3†], Farzaneh Jalalypour[4§], Canan Atilgan[4], Ozan Topcu[2], Nihal Karakas[2,5*], Ozge Sensoy[6,7*]

[1]Graduate School of Engineering and Natural Sciences, Istanbul Medipol University, Istanbul, Turkey; [2]Medical Biology and Genetics Program, Graduate School for Health Sciences, Istanbul Medipol University, Istanbul, Turkey; [3]Cancer Research Center, Institute for Health Sciences and Technologies (SABITA), Istanbul Medipol University, Istanbul, Turkey; [4]Faculty of Engineering and Natural Sciences, Sabanci University, Istanbul, Turkey; [5]Department of Medical Biology, International School of Medicine, Istanbul Medipol University, Istanbul, Turkey; [6]Department of Computer Engineering, School of Engineering and Natural Sciences, Istanbul Medipol University, Istanbul, Turkey; [7]Regenerative and Restorative Medicine Research Center (REMER), Institute for Health Sciences and Technologies (SABITA), Istanbul Medipol University, Istanbul, Turkey

*For correspondence:
nkarakas@medipol.edu.tr (NK);
osensoy@medipol.edu.tr (OS)

[†]These authors contributed equally to this work

Present address: [‡]Molecular Simulations and Design Group, Max Planck Institute for Dynamics of Complex Technical Systems, Magdeburg, Germany; [§]Department of Applied Physics, Science for Life Laboratory, KTH Royal Institute of Technology, Stockholm, Sweden

Competing interest: The authors declare that no competing interests exist.

**Abstract** Undruggability of RAS proteins has necessitated alternative strategies for the development of effective inhibitors. In this respect, phosphorylation has recently come into prominence as this reversible post-translational modification attenuates sensitivity of RAS towards RAF. As such, in this study, we set out to unveil the impact of phosphorylation on dynamics of HRAS[WT] and aim to invoke similar behavior in HRAS[G12D] mutant by means of small therapeutic molecules. To this end, we performed molecular dynamics (MD) simulations using phosphorylated HRAS and showed that phosphorylation of Y32 distorted Switch I, hence the RAS/RAF interface. Consequently, we targeted Switch I in HRAS[G12D] by means of approved therapeutic molecules and showed that the ligands enabled detachment of Switch I from the nucleotide-binding pocket. Moreover, we demonstrated that displacement of Switch I from the nucleotide-binding pocket was energetically more favorable in the presence of the ligand. Importantly, we verified computational findings in vitro where HRAS[G12D]/RAF interaction was prevented by the ligand in HEK293T cells that expressed HRAS[G12D] mutant protein. Therefore, these findings suggest that targeting Switch I, hence making Y32 accessible might open up new avenues in future drug discovery strategies that target mutant RAS proteins.

## Editor's evaluation

This study employs extensive MD simulations to probe the effect of phosphorylation of a tyrosine residue on the conformational ensemble of Ras GTPase. The insights form the basis for a screen of small molecule(s) that disrupt interaction with its target Raf kinase, and predictions are tested experimentally. Overall, the integrated approach is of interest to a wide range of biochemists and protein scientists and could potentially be used to modulate the activities of other proteins.

## Introduction

The RAS gene family translates into four proteins, namely HRAS, NRAS, KRAS4A, and KRAS4B, that control mitogen-activated protein kinase (MAPK), phosphatidylinositol 3-kinase (PI3K), and Ras-like (RAL) pathways (*Barbacid, 1987*; *Malumbres and Barbacid, 2003*; *Lu et al., 2016a*; *Khan et al.,*

*2019*; *Duffy and Crown, 2021*; *Simanshu et al., 2017*; *Ferro and Trabalzini, 2010*; *De Luca et al., 2012*; *Young et al., 2013*; *Knight and Irving, 2014*). These small G proteins act as a binary switch as the activation of the protein is modulated by two types of nucleotides, namely, guanosine-triphosphate (GTP) and guanosine-diphosphate (GDP). The exchange of GDP for GTP is maintained by guanine exchange factors (GEFs) which, in turn, activates the RAS protein (*Downward, 1990*; *Grand and Owen, 1991*; *Bourne et al., 1991*; *Wittinghofer and Pai, 1991*; *Lowy et al., 1991*; *Wittinghofer and Vetter, 2011*; *Takai et al., 2001*; *Lamontanara et al., 2014*; *Vetter and Wittinghofer, 2001*; *Lu et al., 2016a*). Consequently, activated RAS proteins can interact with their downstream effectors, thus initiating cellular signaling pathways (*Vetter and Wittinghofer, 2001*; *Cherfils and Zeghouf, 2013*; *Geyer and Wittinghofer, 1997*; *Lu et al., 2016a*). Unlike GEFs, GTPase-activating-proteins

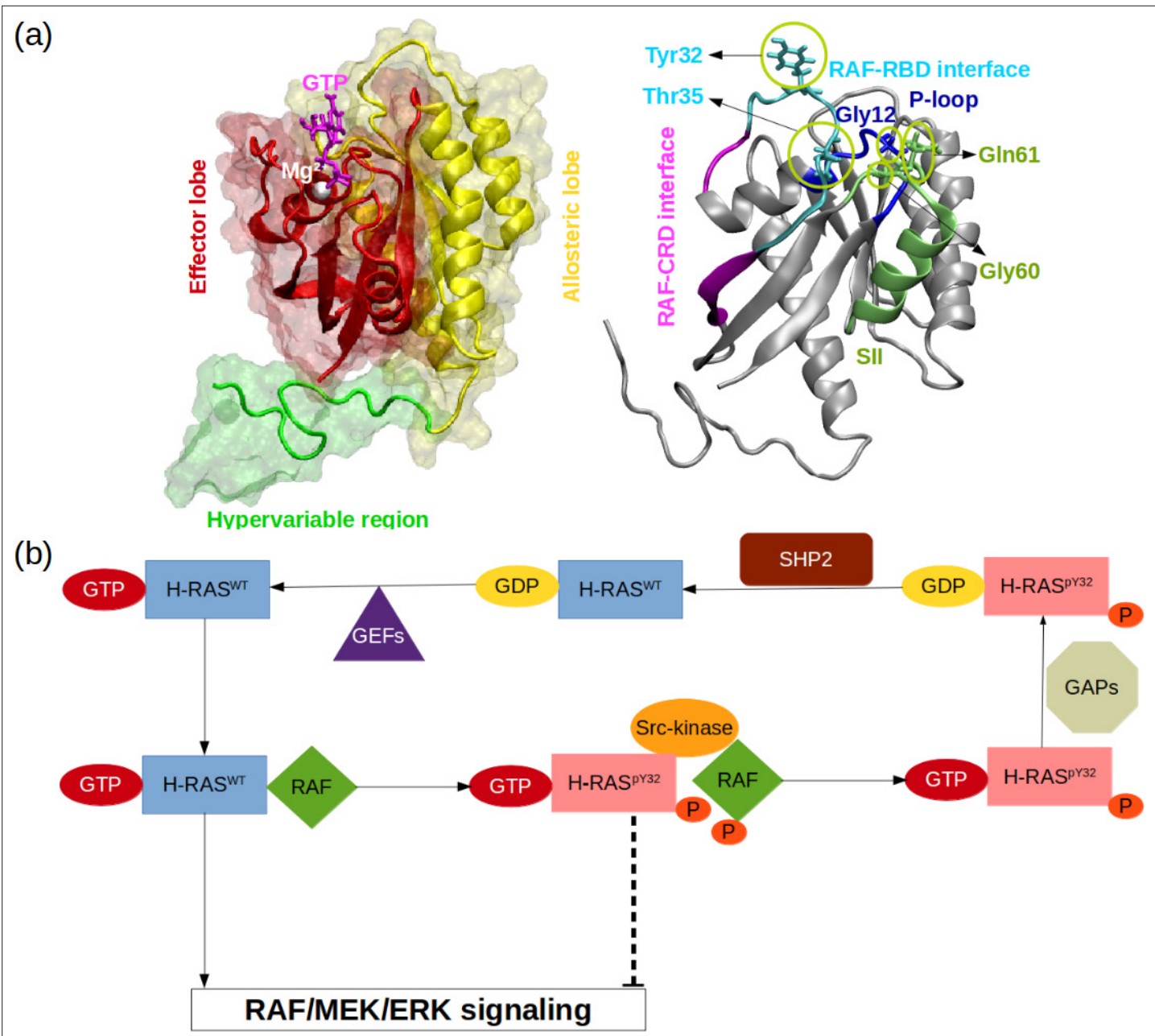

**Figure 1.** RAS phosphorylation/dephosphorylation cycle. (**a**) Important residues/regions that play pivotal role in RAS function are shown. (**b**) A schematic that illustrates the impact of tyrosyl phosphorylation on the GTPase cycle of HRAS. The tyrosyl phosphorylation at the 32<sup>nd</sup> position, which is mediated by Src kinase, causes impairment of RAF binding, thus terminating RAF/MEK/ERK signaling pathway as long as the phosphoryl group of Y32 is not detached by SHP2 *Bunda et al., 2014*.

(GAPs) accelerate the intrinsic GTPase activity of RAS, which provides a control mechanism for precise termination of respective signaling pathways (*Wittinghofer et al., 1997*; *Lu et al., 2016a*).

RAS proteins are made up of two domains, namely, G domain (residues 1–172) and hypervariable region (173–188 or –189) (*O'Bryan, 2019*; *Khan et al., 2020*, *Figure 1A*). The G domain consists of effector (residues 1–86) and allosteric lobes (residues 87–172). The former, which is the invariant region, harbors the P-loop (residues 10–17), Switch I (30-38), and Switch II (59-76) regions, the last two of which adopt different conformational states depending on the type of the nucleotide (*Wang et al., 2021*). In particular, Switch I/II can be found in either open or closed conformation, both of which are described depending on the position of the domain with respect to the nucleotide-binding pocket. In the open conformation, Switch I/II is far from the nucleotide-binding pocket, whereas it is closer in the closed conformation. Importantly, the former prevents effector binding while the latter favors it. Moreover, it has been also shown that Switch II becomes less stable upon effector binding, which presumably allows RAS to cycle between catalytically incompetent and competent states in a timely manner that is important for maintaining the cell homeostasis (*Johnson and Mattos, 2013*; *Khan et al., 2020*).

Since RAS proteins are involved in signaling pathways, which are responsible for cell growth, differentiation, and proliferation, mutations, which are frequently found at the 12th, 13th, and 61st residues (*Prior et al., 2020*; *Prior et al., 2012*, cause several cancer types *Holderfield et al., 2014*; *Eser et al., 2014*; *Prior et al., 2012*; *Stephen et al., 2014*; *McCormick, 2015a*; *McCormick, 2015b*; *Krens et al., 2010*; *Lu et al., 2016a* as a result of attenuated GTP hydrolysis and increased nucleotide exchange rate *Vigil et al., 2010*). For instance, HRAS$^{G12D}$ was shown to be the dominant mutant in ductal carcinoma (*Myers et al., 2016*) caused resistance to erlotinib, which is used as an epidermal growth factor receptor tyrosine kinase inhibitor (*Hah et al., 2014*), in head and neck squamous carcinoma. As such, RAS proteins have been standing as hot targets in drug discovery studies which are focused on the development of therapeutics against cancer.

In spite of extensive efforts that have been made to develop RAS inhibitors, no molecules have yet been approved for clinical use (*Canon et al., 2019*; *Duffy and Crown, 2021*). In these studies, the mutant RAS was targeted directly or in combination with other proteins including SOS, tyrosine kinase, SHP2, and RAF. Also, except KRAS$^{G12C}$ mutant, the GTP-bound state of mutants has been targeted as they either lose their intrinsic or GAP-mediated GTPase activity. However, the intrinsic GTPase activity of KRAS$^{G12C}$ is relatively higher than the other mutants which enables targeting in its GDP-bound state (*Moore et al., 2020*).

The undruggability of RAS proteins arises from lack of deep binding pockets on the surface of the protein and also picomolar affinity of the endogenous ligands which hinders development of competitive inhibitors (*Gysin et al., 2011*; *Ledford, 2015*; *Cox et al., 2014*; *Milroy and Ottmann, 2014*). Therefore, much attention has been focused on the discovery of allosteric sites that can regulate the function of the protein (*Buhrman et al., 2010*; *Ostrem et al., 2013*; *Fetics et al., 2015*; *Johnson et al., 2017*; *McCarthy et al., 2019*; *Khan et al., 2022*).

Importantly, it is well-established that the function of the protein is modulated by post-translational modifications. In particular, phosphorylation/dephosphorylation can be given as an example, which is controlled by Src-kinase and Src homology region 2 domain-containing phosphatase-2 (SHP2), respectively (*Figure 1B*). It has been shown that phosphorylation of the tyrosine at the 32nd position by Src-kinase attenuated RAF binding to HRAS and NRAS while elevating intrinsic GTPase activity of the proteins (*Bunda et al., 2014*, *Figure 1B*). Furthermore, recently, Kano et al. have implied that Src-kinase phosphorylated tyrosine residues at the 32nd and 64th positions of KRAS4B isoform changed conformation of Switch I and II. Consequently, this led to a decrease in intrinsic GTPase activity, thus maintaining KRAS4B in the GTP-bound state. Interestingly, phosphorylated and GTP-bound KRAS4B was shown not to bind RAF, thus leaving the protein in the dark state (*Kano et al., 2019*). In the same study, it was also shown that if phosphoryl groups were removed by SHP2, then GTP-bound KRAS4B could interact with RAF and initiate signaling pathways through MAPK (*Kano et al., 2019*). Notably, it was shown that deletion or inhibition of SHP2 could slow down tumor progression, but remaining insufficient for tumor regression (*Ruess et al., 2018*). Collectively, these findings suggest that mimicking dynamics invoked by phosphorylation might provide an alternative strategy for inhibiting mutant RAS/RAF interaction.

In this study, we set out to investigate the impact of phosphorylation on the structure and dynamics of HRAS[WT] by performing atomistic MD simulations. Comparison of the trajectory pertaining to the phosphorylated RAS with trajectories of GTP-bound HRAS[WT] and HRAS[G12D] showed that phosphorylation of Y32 increased the flexibility of RAF-RBD interface compared to the mutant and pushed Switch I, in particular Y32, out of the nucleotide-binding pocket. Considering the fact that, exposed Y32 precluded RAF binding, we searched for molecules that could evoke similar rearrangements in HRAS[G12D]. To this end, we carried out virtual screening by using therapeutically-approved molecules deposited in DrugBank (*Wishart et al., 2018*; *Law et al., 2014*; *Knox et al., 2011*; *Wishart et al., 2008*; *Wishart et al., 2006*, BindingDB *Gilson et al., 2016*; *Liu et al., 2007*; *Chen et al., 2001b*; *Chen et al., 2001a*; *Chen et al., 2002*, DrugCentral *Ursu et al., 2019*; *Ursu et al., 2017*, and NCGC *Huang et al., 2011*). The impact of ligands on the structure and dynamics of HRAS[G12D] mutant was examined using molecular dynamics simulations. We showed that cerubidine, tranilast, nilotinib, and epirubicin could induce similar dynamics and structural changes which were seen in the phosphorylated RAS protein. Moreover, we also calculated the energy required for pushing Switch I out of the nucleotide-binding pocket in the absence/presence of one of the successful ligands, namely cerubidine, using perturb-scan-pull (PSP) method *Jalalypour et al., 2020* and showed that less energy was required for displacement of Switch I in the presence of the ligand. Importantly, we also tested the activity of cerubidine in preventing RAS/RAF interaction using immunoprecipitation assays and verified computational findings. Therefore, these results suggest that Y32 detachment from the nucleotide-binding pocket might be used as an alternative strategy for targeting mutant RAS proteins.

## Results

### Phosphorylation impacts the flexibility of Y32 and RAF-RBD/RAS interface residues

The comparison of RMSF profiles showed remarkable differences in the fluctuation patterns of certain residues/domains among wild-type, phosphorylated, and mutant protein. We showed that phosphorylation increased the flexibility of Y32 as a result of repulsion between negatively charged phosphate and GTP and also increased the flexibility of the residues that are involved in the RAF-RBD interaction interface compared to the mutant as shown in *Table 1*. Since RAF-RBD plays a major role in formation of high-affinity RAS-RAF complex, increased flexibility presumably precludes RAF binding. On the other hand, phosphorylation did not impact the RAF-CRD interface, which has been shown to play an important role in anchoring RAF to the membrane (*Travers et al., 2018*) as revealed by NMR and mutagenesis studies (*Drugan et al., 1996*). It is important to note that the difference seen in the flexibility of Y32 and RAF-RBD interface residues between HRAS[pY32] and HRAS[G12D] could not be observed for G60 and Q61, although the two residues could reach relatively higher RMSF values in the phosphorylated RAS (see the standard errors in *Table 1*). This, in turn, which might cause perturbation of the interface formed with RAF-CRD and modulate GAP-mediated GTPase activity of RAS, respectively.

**Table 1.** The backbone RMSF values of key regions/residues pertaining to HRAS[WT], HRAS[pY32], and HRAS[G12D].

| Residue/Region-RMSF (Å) | HRAS[WT] | HRAS[pY32] | HRAS[G12D] |
|---|---|---|---|
| Y32 | 1.6 ± 0.2 | 1.5± 0.2 | 0.9± 0.1 |
| RAF-RBD interface residues | 1.2± 0.2 | 1.3± 0.2 | 0.8± 0.1 |
| RAF-CRD interface residues | 0.8± 0.1 | 0.9± 0.2 | 0.7± 0.03 |
| G60 | 1.0± 0.3 | 1.4± 0.5 | 1.1± 0.2 |
| Q61 | 1.2± 0.4 | 1.6± 0.4 | 1.5± 0.3 |

## Phosphorylation pushes Switch I and Y32 out of the nucleotide-binding pocket of RAS

As shown in *Table 1*, the flexibility of residues, which interact with RAF-CRD domain, increased upon phosphorylation compared to HRAS[G12D]. Since these residues surround the Switch I domain, we sought to investigate whether the opening of the nucleotide-binding pocket was impacted by measuring the distance between Cα atoms of the G/D12 and P34 residues throughout the trajectories. We showed that phosphorylation pushed Switch I out of the binding pocket as the distance between G12 and P34 residues increased compared to the mutant protein (*Figure 2A*, *Figure 2—figure supplement 1A-C*). Consequently, this makes the nucleotide-binding pocket more accessible to waters, as evident from the number of waters measured within 5 Å distance of GTP: 93.60±0.20, 100.30±0.20, and 72.68±0.17 for HRAS[WT], HRAS[pY32], and HRAS[G12D], respectively. Considering rearrangements occurring around the nucleotide-binding pocket and that GAP binding involves interaction with both Switch I and II, it can be said that phosphorylation triggers rearrangement around the pocket of RAS that prepares it for binding to GAP, thus, modulating the GTPase activity of the protein. Indeed, it was shown that GTPase activity of RAS increased upon phosphorylation (*Bunda et al., 2014*).

Having observed phosphorylation-induced modulation in the flexibility of Y32, we also examined the positioning of the residue by measuring the distance between the side-chain oxygen of Y32 and Pγ atom of GTP (see *Figure 2—figure supplement 1D,E and F*). We showed that Y32 formed a hydrogen bond with the Pγ atom of GTP in both HRAS[WT] and HRAS[G12D] which stabilized the residue in the vicinity of the nucleotide-binding pocket (*Figure 2E*). However, the hydrogen bond was not formed in HRAS[pY32], and Y32 was positioned far from the pocket, thus making it exposed to the environment, as evidenced by relatively longer distances measured (*Figure 2B and D*). In addition to the position, we also explored orientational preference of Y32 with respect to the nucleotide-binding pocket by measuring dihedral angles pertaining to backbone and side-chains of Y32, namely $\phi/\psi$ and $\chi_1/\chi_2$ angles. There was no remarkable difference in backbone dihedrals and $\chi_1$, whereas $\chi_2$ angle distribution was different among the systems studied. Specifically, Y32 displayed two peaks at –100 to –90° and 80–90° in the phosphorylated RAS, whereas it adopted 60–70° in the mutant and wild-type HRAS (*Figure 3A* and *Figure 3—figure supplement 1*). It is important to point that Y32 adopted 80° in the crystal structure of allosteric inhibitor-bound KRAS4B[G12D] (PDB ID:6WGN) *Zhang et al., 2020*, where the residue was exposed and far from the nucleotide-binding pocket as the distance measured between the side-chain of Y32 and Pγ atom of GTP was 16 Å.

Herein, it is important to mention that exposed conformation of Y32 was not observed in the trajectories pertaining to RAF-RBD-bound HRAS[WT] as shown in our earlier study (*Ilter and Sensoy, 2019*). Therefore, this finding suggests that exposure of Y32 might occlude the interaction interface formed between RAS and RAF-RBD.

## Global dynamics reveals a possible binding site near the nucleotide-binding pocket in HRAS[G12D]

Besides local analysis, the collective dynamic properties of the systems were also explored by calculating the principal components of their global motions. To do so, trajectories of HRAS[WT], HRAS[G12D], and HRAS[pY32] were projected along their first five eigenvectors, which reflect *ca.* 50% of the overall dynamics *Petrone and Pande, 2006* (see *Figure 4—figure supplement 1*), and compared to each other to investigate global conformational rearrangements induced by phosphorylation and mutation. Consequently, it was shown that the dynamics of Switch I domain, in particular RAF-RBD interaction interface, could be altered in the mutant, albeit to a lesser extent, compared to both HRAS[WT], HRAS[pY32]. It is also important to mention that the contribution of Y32 to the first five eigenvectors was higher in HRAS[pY32] (3.14 Å) than in the mutant protein (2.87 Å) as seen in the RMSF values.

ON the other hand, the contribution of Switch II, which harbors both G60 and Q61, to the overall dynamics was higher in HRAS[WT] and HRAS[pY32] than the mutant protein (*Figure 4A*). Q61 was positioned closer to the nucleotide-binding pocket in the mutant HRAS than in phosphorylated HRAS whereas G60 could sample longer distances than that was sampled by both HRAS[pY32] and HRAS[WT] at their maximum probability (*Figure 4C and D* and *Figure 4—figure supplement 2*).

Having observed flexibility at the RAF-RBD interface in the mutant, we set out to investigate if the site can be considered as a possible binding pocket that can accommodate small molecules to modulate the dynamics of Switch I. To do so, we clustered the trajectory of the mutant HRAS by

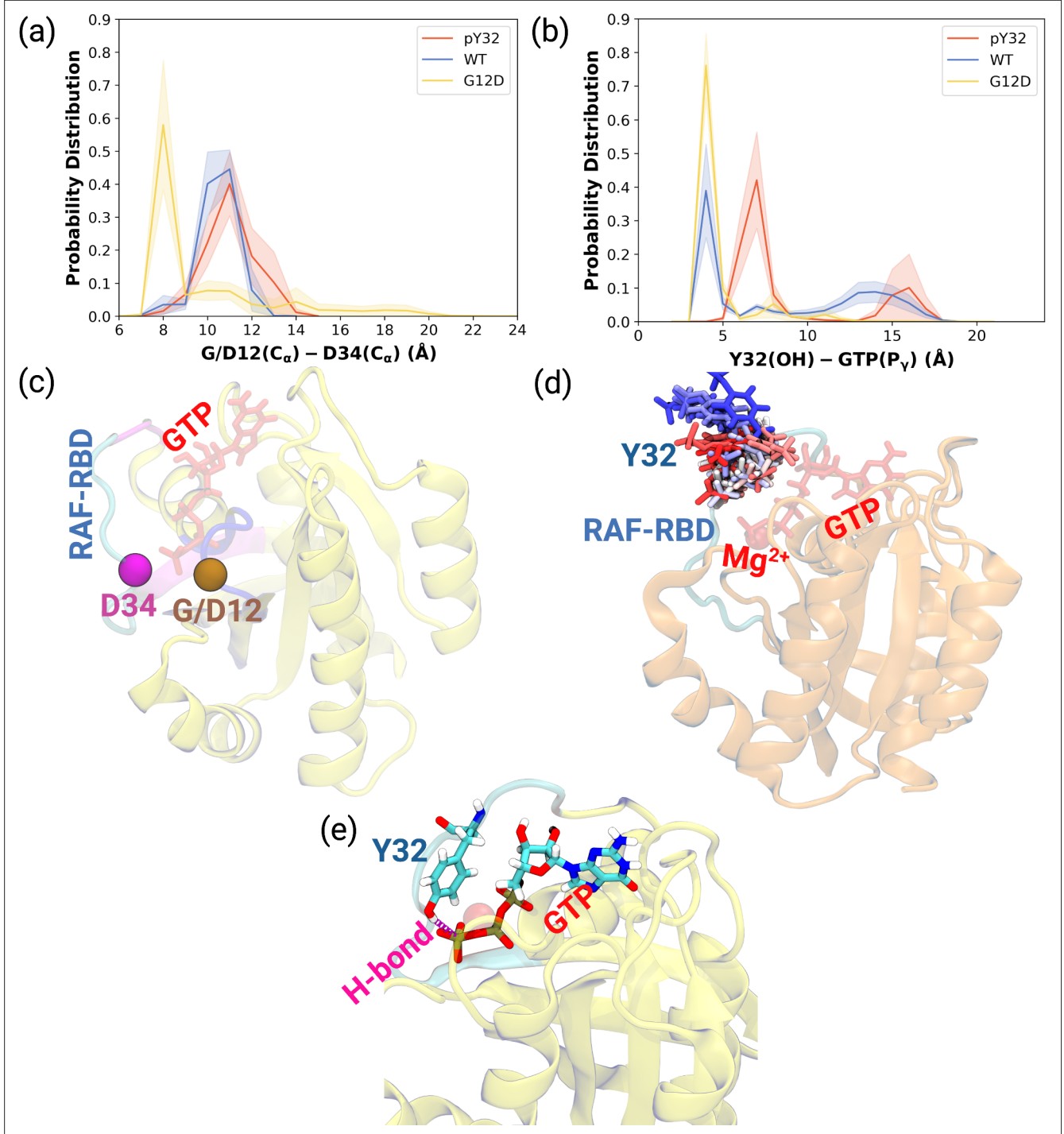

**Figure 2.** Probability distance distributions pertaining to nucleotide binding pocket and respective schematic representation of the residues used for the calculations. (**a**) The normalized probability density distribution of the distance measured between the Cα atom of 12th 34th residues. The standard error (SE) of the measurements for HRAS[pY32], HRAS[WT], and HRAS[G12D] are measured as 0.01, 0.004, and 0.01 Å, respectively. (**b**) The normalized probability distribution of the distance between the side-chain oxygen atom of Y32 and Pγ of GTP. The SE of the measurements for HRAS[pY32], HRAS[WT], and HRAS[G12D] are calculated as 0.02, 0.02, and 0.003 Å, respectively. (**c**) Cα atoms of 12th and 34th residues are shown on the crystal structure of HRAS[WT] (PDB ID: 5P21) in vdW representation and colored with ocher and purple, respectively, whereas GTP is shown in the licorice and colored with red. (**d**) The orientational dynamics of Y32 in the HRAS[pY32] trajectory. (**e**) The H-bond formed between side-chain of Y32 and Pγ of GTP in HRAS[G12D] is shown in purple.

The online version of this article includes the following figure supplement(s) for figure 2:

*Figure 2 continued on next page*

*Figure 2 continued*

**Figure supplement 1.** The timeline interatomic distances between Cα atoms of G/D12 and D34 throughout (**a**) HRAS$^{WT}$, (**b**) HRAS$^{G12D}$, and (**c**) HRAS$^{pY32}$ trajectories.

considering probability density distributions of distances between the (i) side-chain oxygen of T35 and Pγ of GTP (ii) backbone amide of G60 and Pγ of GTP, which represent different conformational states of the nucleotide-binding pocket, according to the structural studies (*Lu et al., 2016b*; *Shima et al., 2010*; *Araki et al., 2011*). There were three states described for T35, labelled as state 1, 2, and 3, each of which sampled distances in the range of 3.0–5.0 Å, 6.0–9.0 Å, and 12.0–16.0 Å, respectively (*Figure 5A* and *Figure 5—figure supplement 1*). Similarly, G60 could also adopt three states, namely state 1, 2, and 3, which corresponds to distance range between 5.0–7.0 Å, 2.0–4.0 Å, and 8.0–9.0 Å, respectively (*Figure 5B*). In light of clustered conformations, the most probable conformation that adopts values pertaining to State 1 in each atom-pair distances was picked up from the trajectory of HRAS$^{G12D}$. The possible binding pockets on the surface of mutant HRAS were identified and evaluated by comparing SiteMap scores. Eventually, the pocket, which had relatively higher SiteScore, enclosure and lower exposure, was selected to be used further (*Table 2*). The binding pocket, which was identified on the selected conformation, was next to Switch I. Considering the fact that this domain (i) includes residues that mediate RAF binding, (ii) acts as a regulator for intrinsic GTPase activity, and (iii) dominates the collective dynamics of the mutant protein, the region was used as the target binding pocket in the subsequent steps of the study.

## Small therapeutic molecules distort the RAF binding interface and pushes Y32 out of the pocket

The pharmacophore groups of the binding site identified on the surface of HRAS$^{G12D}$ were modeled with respect to both geometrical and chemical properties of residues 29–34. DrugBank (*Wishart et al., 2018*; *Law et al., 2014*; *Knox et al., 2011*; *Wishart et al., 2008*; *Wishart et al., 2006*, Drug-Central *Ursu et al., 2019*; *Ursu et al., 2017*, BindingDB *Gilson et al., 2016*; *Liu et al., 2007*; *Chen et al., 2001b*; *Chen et al., 2001a*; *Chen et al., 2002*, and NCGC *Huang et al., 2011*) databases

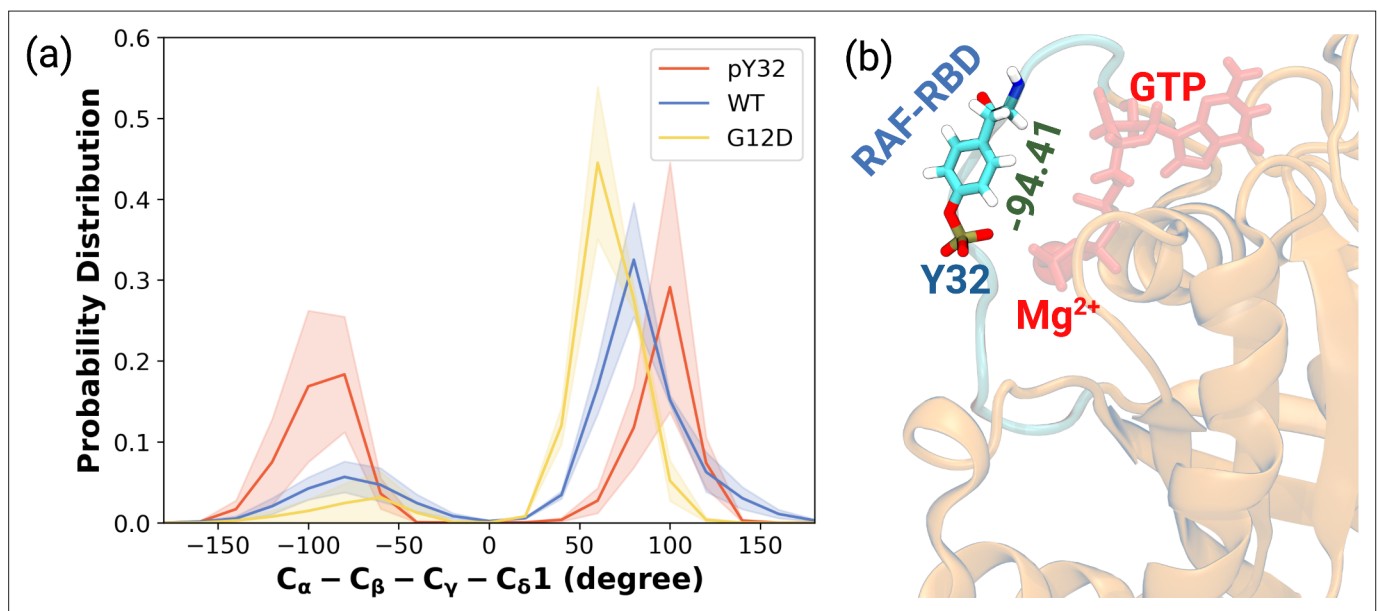

**Figure 3.** The normalized probability density distribution plots pertaining to chi2 angle of residue Y32 along with respective snapshot.
 (**a**) The normalized probability density distribution of the measured $\chi_2$ angles of HRAS$^{WT}$, HRAS$^{G12D}$, and HRAS$^{pY32}$. The SE of $\chi_2$ pertaining to the phosphorylated, wild-type, and mutant systems are 0.44, 0.35, and 0.09, respectively. (**b**) A representative exposed state of Y32 obtained from the trajectory of the phosphorylated system.

The online version of this article includes the following figure supplement(s) for figure 3:

**Figure supplement 1.** The timeline measured $\chi_2$ angles of Y32 throughout (**a**) HRAS$^{WT}$, (**b**) HRAS$^{G12D}$, and (**c**) HRAS$^{pY32}$ trajectories.

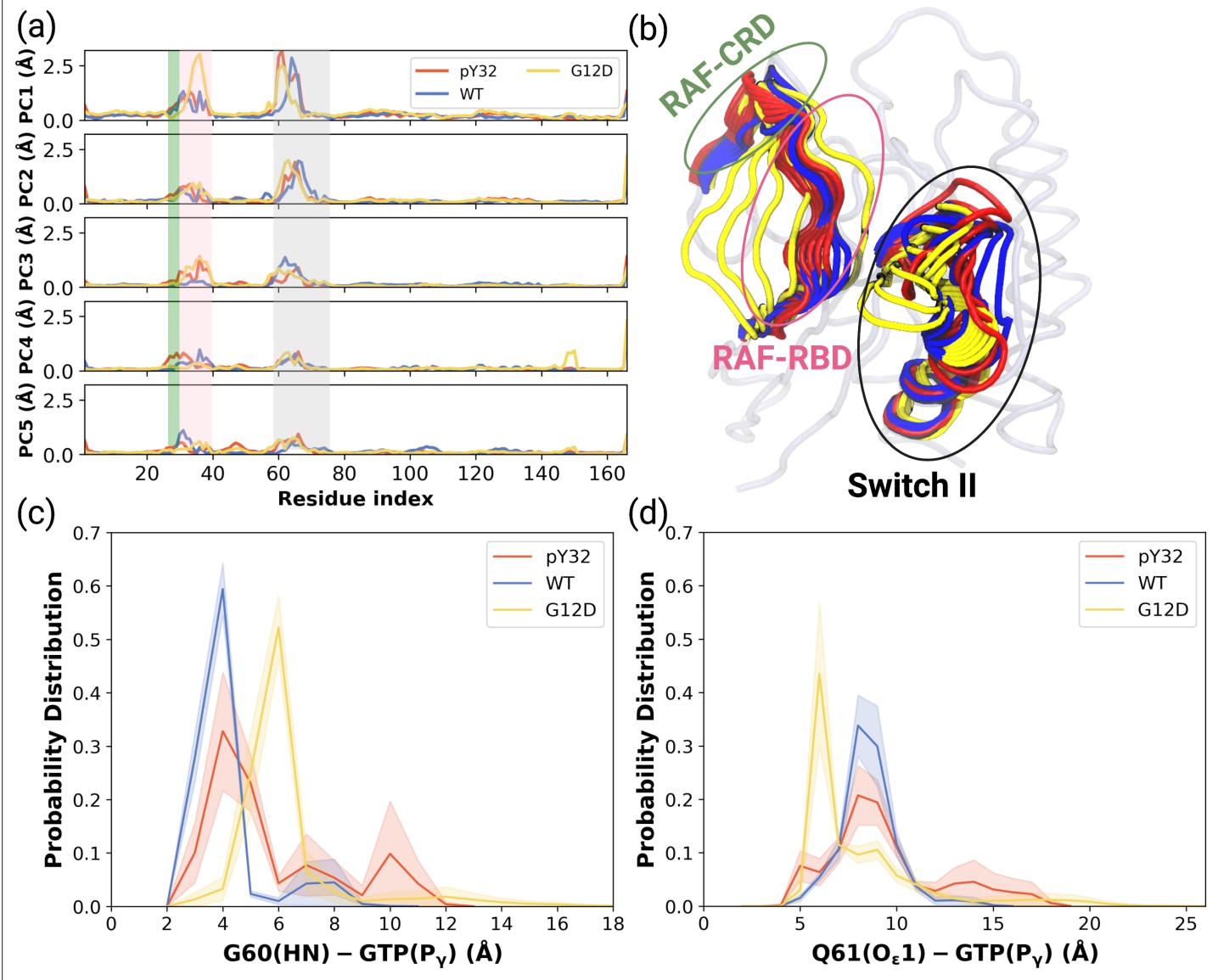

**Figure 4.** PCA analysis and Distance Probability Distributions pertaining to G60/Q61 and the nucleotide. (**a**) Fluctuation of Cα atoms pertaining to HRAS^WT, HRAS^G12D, and HRAS^pY32 along the first five eigenvectors. The RAF-CRD & -RBD interaction interfaces, as well as Switch II, are shaded in the green, pink, and black rectangles, respectively. The eigen RMSF of Y32 pertaining to the phosphorylation system is pointed out by a dark violate bead. (**b**) The projected trajectories of the systems studied along with the first principal component, where the thickness of the ribbons are correlated the contribution of domain to the collective dynamics. The probability density distribution of the distance between (**c**) the backbone amide of G60 and Pγ of GTP is shown, and (**d**) the side-chain oxygen of Q61 and Pγ of GTP is shown. The SE of the distance between G60 and GTP pertaining to the phosphorylated, wild-type, and mutant systems are 0.01, 0.01, and 0.004 Å, respectively. The SE of the distance between Q61 and GTP pertaining to the phosphorylated, wild-type, and mutat systems are 0.02, 0.01, and 0.01, respectively.

The online version of this article includes the following figure supplement(s) for figure 4:

**Figure supplement 1.** The cumulative contribution of the principal components pertaining to the (**a**) phosphorylated, (**b**) wild-type, and (**c**) G12D mutant systems to the overall dynamics.

**Figure supplement 2.** The timeline of the measured distances between HN of G60 and Pγ of GTP for (**a**) wild-type, (**b**) mutant, and (**c**) phosphorylated systems.

were searched for molecules that could contain at least three features of the modeled pharmacophores and have molecular weight lower than 550 kDa. A total of 4292 molecules was retrieved from the databases (**Figure 6A**). Then, these molecules were docked to the identified binding pocket on the surface of HRAS^G12D and ligands were evaluated with respect to their spatial organization around

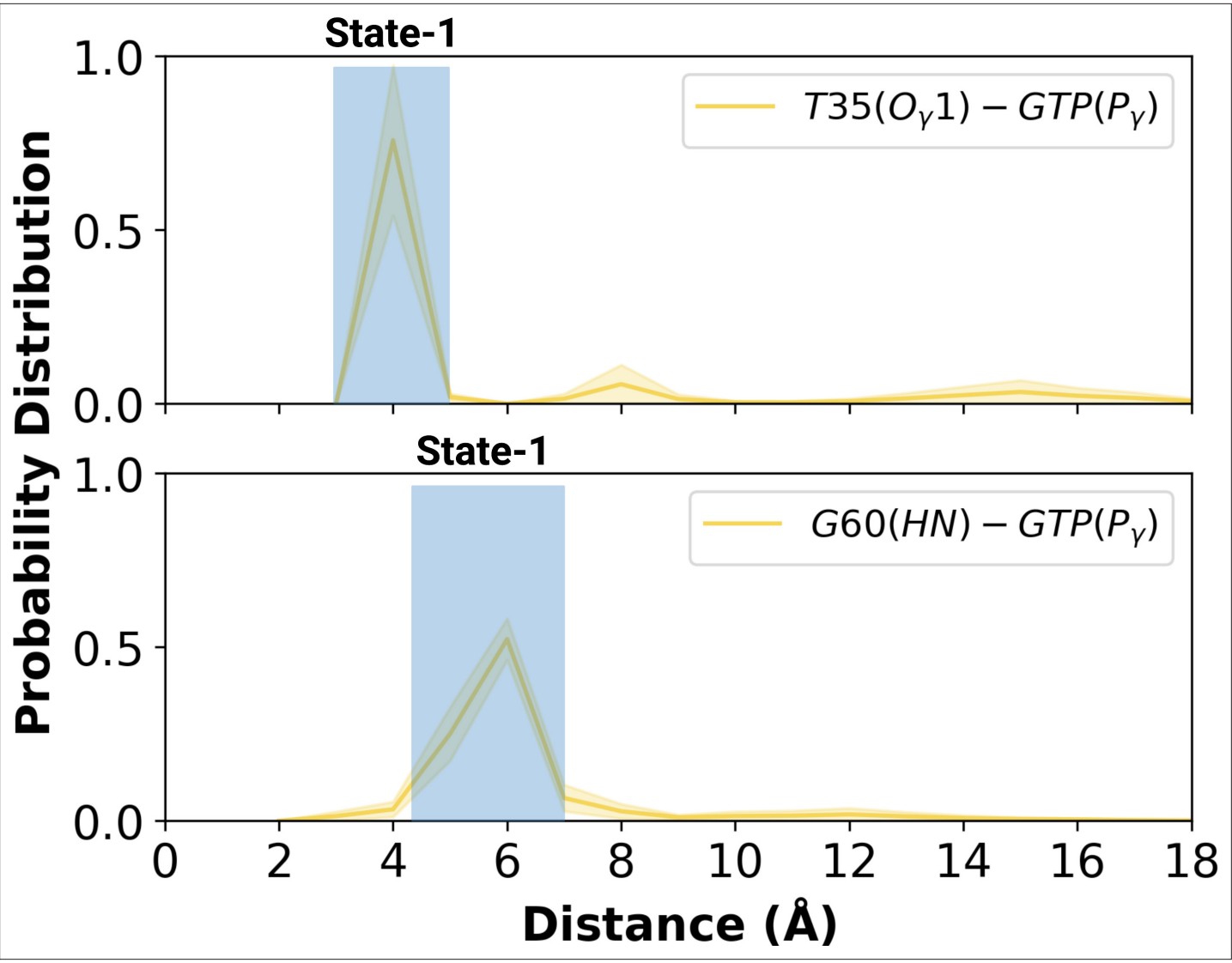

**Figure 5.** Probability distribution plot pertaining to distance measured between T35 (Ogamma)-GTP(Pgamma) and G60 (HN)-GTP (Pgamma) for the most populated conformational state of mutant H-RAS.

The normalized probability density distribution of the measured distance between the side-chain oxygen atom of T35 and Pγ of GTP, and the backbone amide of G60 and Pγ of GTP in the mutant system, where the sampling range for calculating the frequency of each interval was adjusted as 1 Å. The SE of the distance between T35 and GTP is calculated as 0.001 Å, whereas that of the distance between G60 and GTP is 0.004 Å.

The online version of this article includes the following figure supplement(s) for figure 5:

**Figure supplement 1.** The (**a**) timeline and (**b**) histogram of the measured distance between O1 of T35 and Pγ of GTP for the mutant system.

**Table 2.** The SiteMap scores of possible pockets found on the surface of the most probable conformation of HRAS[G12D].

| SiteScore | Size | DScore | Volume | Exposure | Enclosure | Contact | Phobic | Philic | Balance | Don/acc |
|-----------|------|--------|---------|----------|-----------|---------|--------|--------|---------|---------|
| 1.028 | 194 | 0.919 | 466.140 | 0.478 | 0.740 | 1.010 | 0.259 | 1.425 | 0.182 | 0.932 |
| 0.701 | 25 | 0.668 | 81.290 | 0.632 | 0.691 | 1.059 | 1.474 | 0.664 | 2.219 | 0.915 |
| 0.656 | 22 | 0.629 | 101.870 | 0.776 | 0.638 | 0.842 | 0.959 | 0.596 | 1.609 | 12.216 |

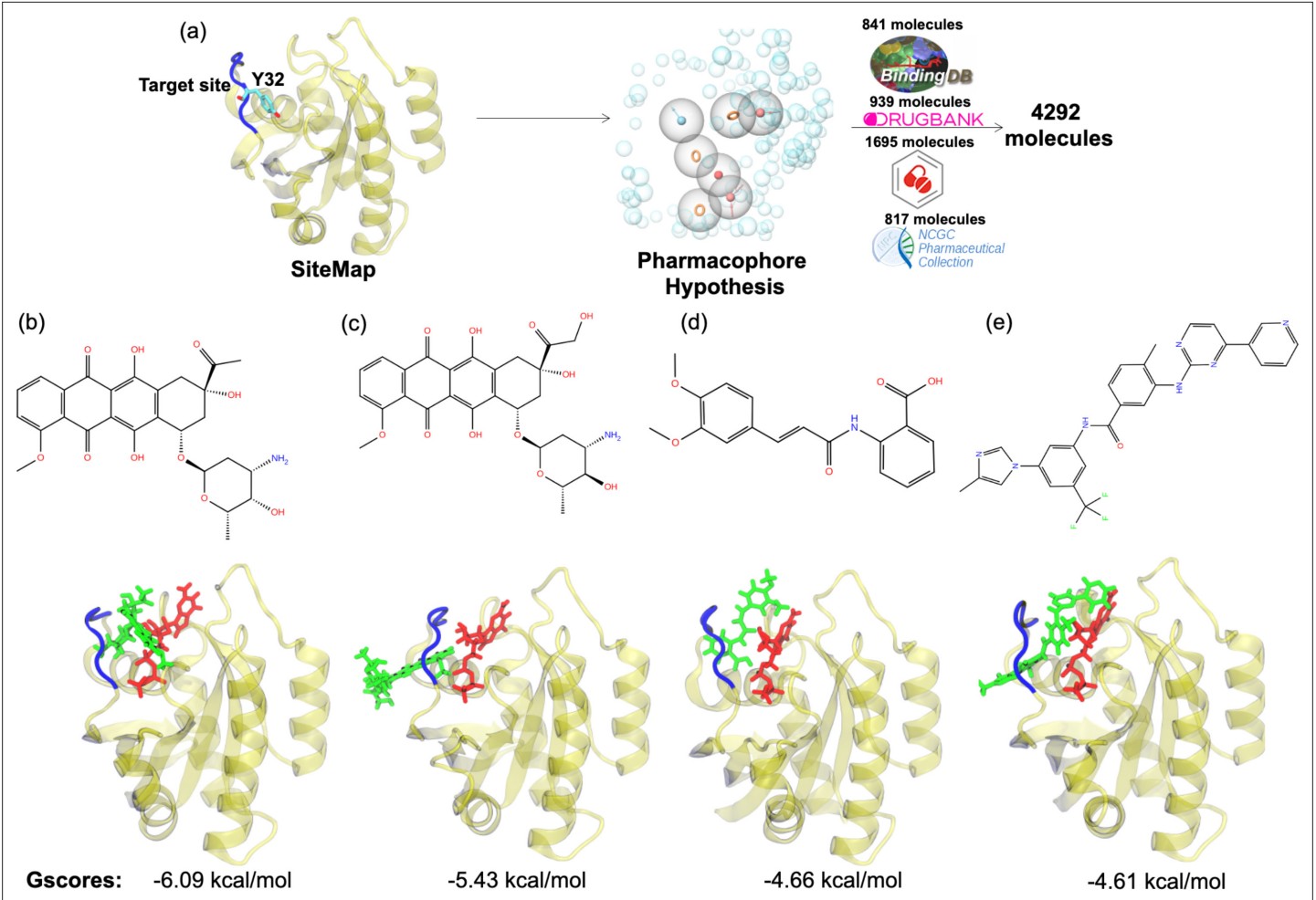

**Figure 6.** Virtual screening scheme used, and candidate molecules achieved along with their chemical structures and docking energy values. (a) A schematic that summarizes the virtual screening workflow done for the identified binding pocket on the most frequently sampled conformation of HRAS[G12D]. The 3D structures and corresponding GScores of (b) cerubidine, (c) epirubicin, (d) tranilast, and (e) nilotinib are shown. GTP is shown in licorice and red.

the nucleotide-binding pocket and GScores, which is a term that is used to score binding poses in Schrodinger. Considering the close interaction observed between Y32 and GTP in HRAS[G12D], we prioritized the ligands, which disrupted interaction between the nucleotide and Y32. The impact of four ligands satisfying this criterion, namely cerubidine, tranilast, nilotinib, and epirubicin, (*Figure 6B-E*) was further tested by performing MD simulations using the ligand-HRAS[G12D] complex (See *Supplementary file 1* for respective simulation times).

The ligand-HRAS[G12D] trajectories were analyzed based on the fluctuation pattern of RAF-RBD, RAF-CRD, and Y32. Moreover, the distances measured between G/D12 and P34 as well as Y32 and GTP were also compared to those of HRAS[G12D] and HRAS[pY32]. In that way, the capability of the ligands in distorting Switch I domain, widening the nucleotide-binding pocket, and displacing Y32 could be investigated. Accordingly, ligands, which could (i) increase the flexibility of RAF-RBD and -CRD interfaces, and (ii) displace Switch I and Y32 from the nucleotide-binding pocket were considered successful in terms of preventing HRAS[G12D]/RAF interaction. We showed that all the ligands, namely cerubidine, nilotinib, tranilast, and epirubicin, considerably increased the flexibility of the RAF-RBD interaction interface (See *Supplementary file 1*) than in HRAS[G12D]. Moreover, the flexibility of Y32, also significantly increased by all the ligands, except nilotinib (*Bunda et al., 2014*; *Kano et al., 2019*, *Supplementary file 1*).

We also examined the wideness of the nucleotide-binding pocket and the positioning of Y32 by measuring the distances between G/D12, respectively. We showed that cerubudine was more likely to

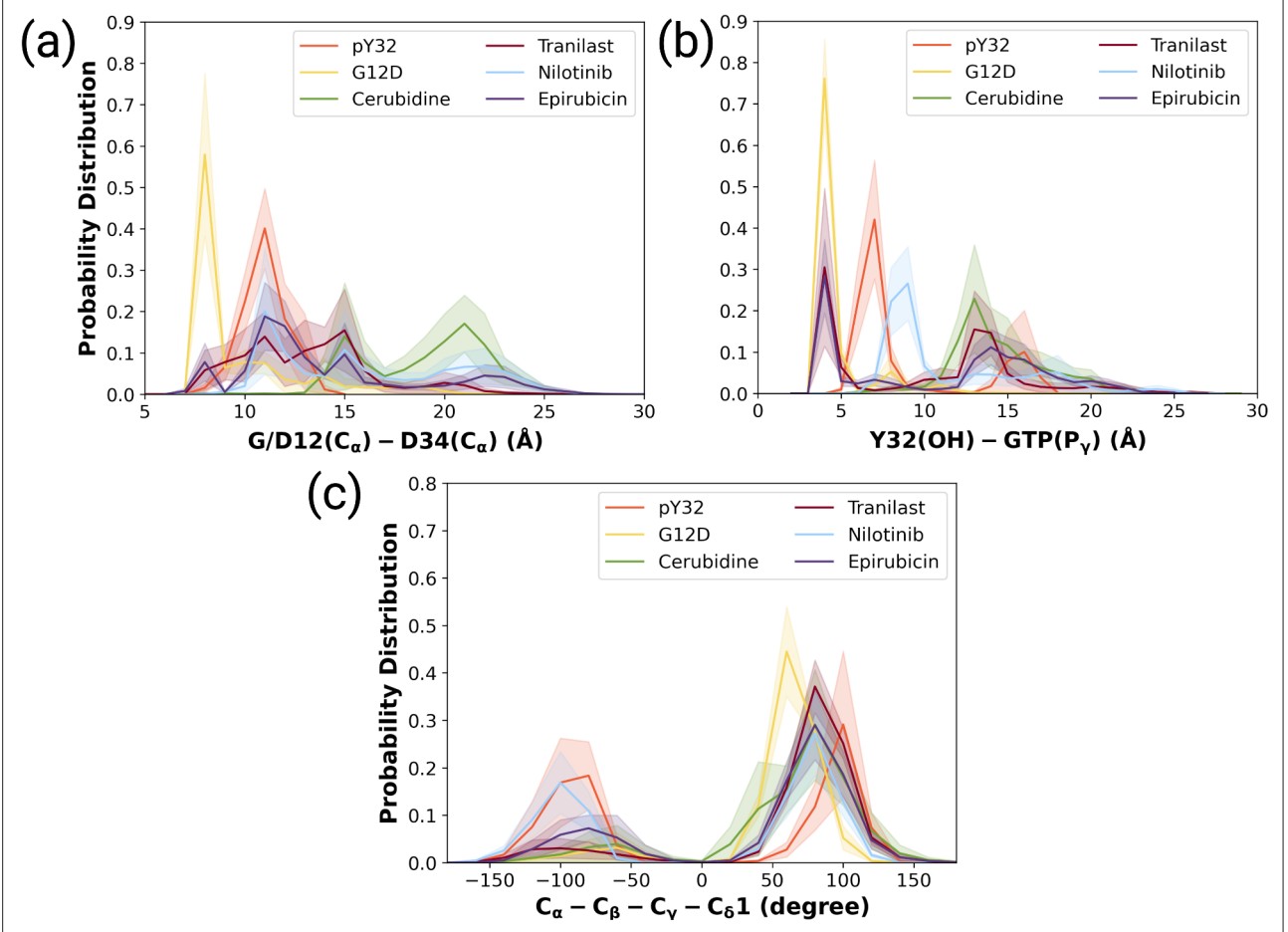

**Figure 7.** The normalized probability density distribution plots pertaining to distances of G/D12 (Calpha)- and D34 (Calpha), Y32 (OH)-GTP (Pgamma) and chi2 angle of Y32 measured in ligand-bound mutant H-RAS trajectories.

The normalized probability density distribution of (**a**) the distance between Cα atoms of G/D12 and P34, (**b**) the distance between the side-chain oxygen atom of Y32 and Pγ of GTP, (**c**) $\chi_2$ in HRAS$^{G12D}$, HRAS$^{pY32}$, and cerubidine-, tranilast-, nilotinib-, and epirubicin-bound HRAS$^{G12D}$. The SE of the G/D12-D34 distance pertaining to the above-mentioned systems, in seriatim, 0.01, 0.01, 0.02, 0.02, 0.03, and 0.03 Å. The SE of the same systems for Y32-GTP distance is 0.003, 0.02, 0.02, 0.03, 0.02, 0.04 Å, respectively. The SE of $\chi_2$ is 0.09, 0.44, 0.37, 0.39, 0.51, and 0.39 degree for the mutant, phosphorylated, cerubidine-, tranilast-, nilotinib, and epirubicin-bound HRAS$^{G12D}$ systems, respectively.

The online version of this article includes the following figure supplement(s) for figure 7:

**Figure supplement 1.** The timeline and histogram of the measured distances between Cα atoms of D12 and D34 for (**a,e**) cerubidine-bound, (**b,f**) tranilast-bound, (**c,g**) nilotinib-bound, and (**d,h**) epirubicin-bound HRAS$^{G12D}$ systems, respectively.

**Figure supplement 2.** The timeline and histogram of the measured $\chi^2$ angles pertaining to Y32 for (**a**) cerubidine-bound, (**b**) tranilast-bound, (**c**) nilotinib-bound, and (**d**) epirubicin-bound HRAS$^{G12D}$ systems.

trigger displacement of Switch I and Y32 away from the nucleotide-binding pocket, whereas the impact of nilotinib and epirubicin was not remarkable (See *Figure 7A and B*). This, in turn, explains accommodation of relatively higher number of waters within the nucleotide-binding pocket in cerubidine-bound HRAS$^{G12D}$ (*Supplementary file 1*). Therefore, it is tempting to suggest that cerubidine can help elevate the intrinsic GTPase activity of the mutant RAS by exposing GTP to water.

Also, Y32 adopted similar $\chi_2$ angle in cerubidine-bound HRAS$^{G12D}$ to that in the phosphorylated RAS (*Figure 7C* and *Figure 7—figure supplement 2*). Considering similarities between ligand-bound HRAS$^{G12D}$ and HRAS$^{pY32}$, cerubidine can be thought to have relatively more potential for preventing HRAS$^{G12D}$/RAF interaction. Therefore, we used cerubidine in the subsequent steps of the study to test our proof-of-concept.

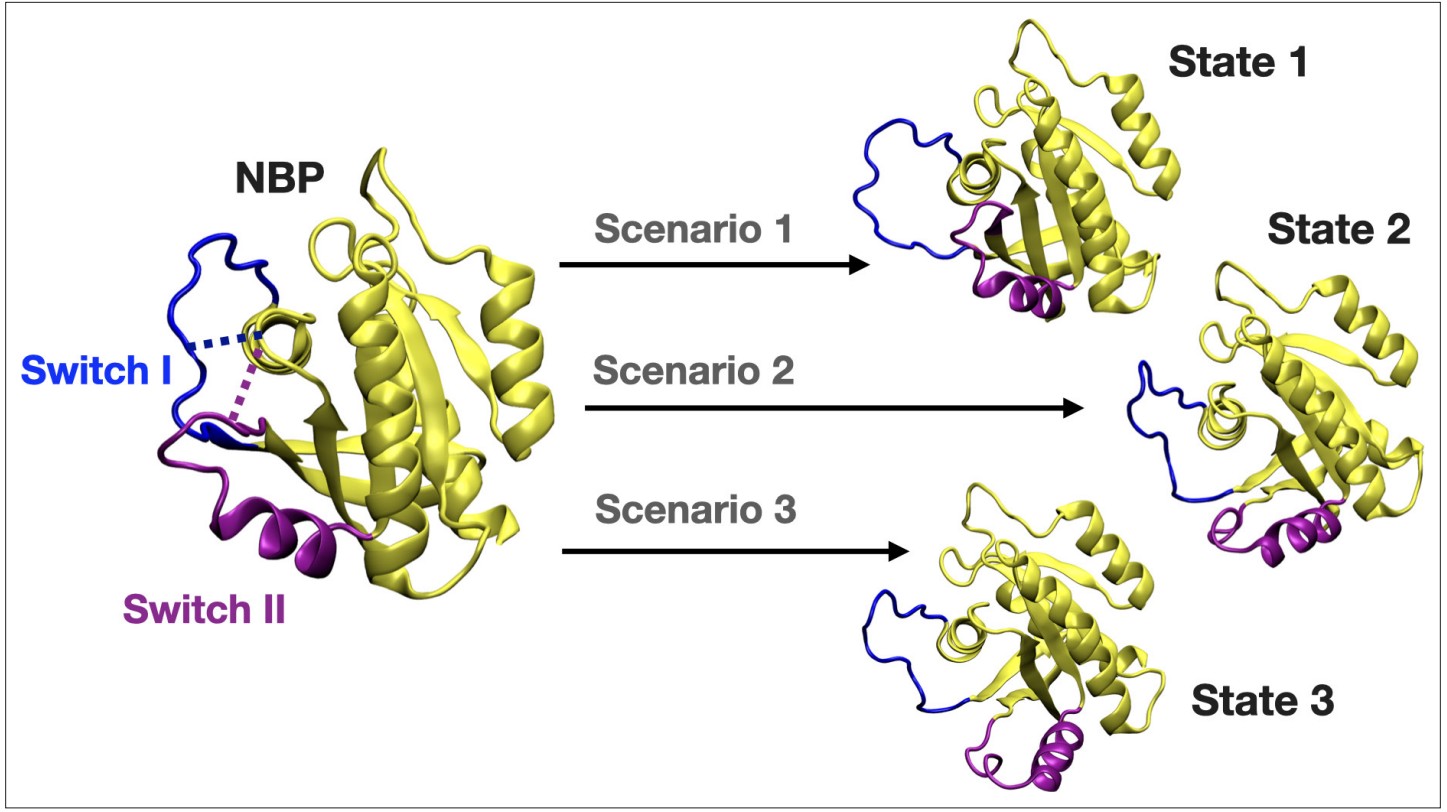

**Figure 8.** A schematic that illustrates the PRS calculations made for examining the transition between the initial and target states. The initial state represents the conformation of the closed-state of Switch I and II. The target state-1 is described as the open state of Switch I (blue) and close state of Switch II (purple). The target state-2 represents the partially open state of Switch I and open state of Switch II. The target state-3 corresponds to open state of Switch I and II.

## Perturb-Scan-Pull method reveals that displacement of Switch I and Y32 in HRAS[G12D] is favored in the presence of cerubidine

We further set out to investigate if displacement of Switch I and Y32 is energetically favorable in the presence of the cerubidine. To do so, we applied perturb-scan-pull (PSP) method (*Jalalypour et al., 2020*), which was developed to investigate conformational transitions in proteins, on cerubidine-bound HRAS[G12D]. In this approach, initial and target states are described and the most possible path for transitioning between the initial and the target state is determined by calculating the overlap between the states. The maximum overlap is thought to give the optimum conformational transition path. To be consistent with the previous analyses, we used the same reaction coordinates, such as the distance between (i) Cα atoms of D12 and P34, and (ii) the backbone amide of G60 and Pγ of GTP, as the reaction coordinates, which, reflected dynamics of Switch I and II, respectively. Accordingly, we described three and two states for Switch I and II, respectively, considering the conformations obtained by clustering of HRAS[G12D] trajectory. Accordingly, for Switch I, if the measured distance between Cα atoms of D12 and P34 is less than 8 Å, Switch I is grouped as in the closed state. On the other hand, when the atom-pair distance is above 16 Å, Switch I grouped as in the open state.

**Table 3.** The results of PRS calculations for the transition between initial and target states.

| Ligand | State | D12-P34 (Å) | G60-GTP (Å) | PRS selected residues | PRS overlap($O^j$) |
|---|---|---|---|---|---|
| Cerubidine[a] | Initial state[b] | 7.7 (closed) | 7.0 (closed) | - | - |
| | Target state-1 | 26.3 (open) | 11.00 (closed) | 34, 35, 33, 32, 37, 36 | 0.74–0.70 |
| | Target state-2 | 13.9 (partially open) | 15.00 (open) | 35, 34, 33, 66, 16, 65 | 0.58–0.50 |
| | Target state-3 | 19.5 (open) | 16.9 (open) | 34, 66, 35, 64, 16, 33 | 0.59–0.50 |

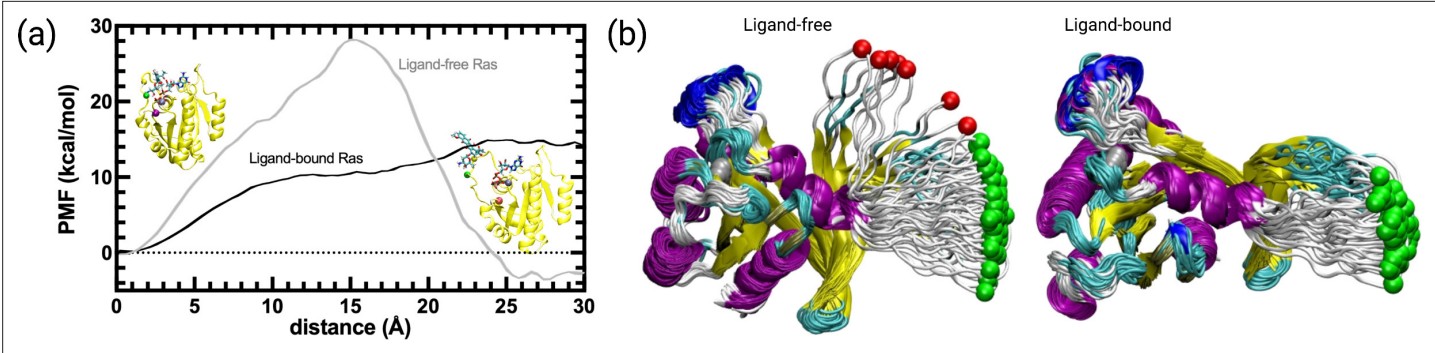

**Figure 9.** PMF along the PSP predicted coordinate with the highest overlap for the transition scenario 1 (Switch I opening motion) as a function of distance. PMF is calculated for the HRAS$^{G12D}$ system in the presence and absence of cerubidine, and each simulation was repeated 70 times. The reaction coordinate refers to the distance covered by the SMD atom (shown as a yellow bead) during the course of the pulling experiments. The initial and final structures of an SMD simulation are illustrated on the left and right sides of the figure, respectively. Yellow bead: Y32; Green bead: P34; Iceblue bead: G60; GTP and Cerubidine: Licorice representation. Errors are indicated by the thickness of the curves and are less than 0.3 kcal/mol and 0.1 kcal/mol in ligand-free and ligand-bound simulations, respectively.

The online version of this article includes the following figure supplement(s) for figure 9:

**Figure supplement 1.** The free energy profile, where the error estimate was calculated by usingbootstrapping.

The distance between 8 and 16 Å is grouped as the partially open state of Switch I. Likewise, we also determined the state of the Switch II by measuring the distance between the backbone amide of G60 and Pγ of GTP. If the distance is above 11 Å, Switch II is grouped as in the open state, if not, in the closed state. In light of these atom-pair distances, the initial state was described as the closed state of both Switch I and II domains, since it was the most frequently sampled conformation in trajectories of the mutant HRAS. As to the target states, we described three such scenarios as shown in *Figure 8*. The target state-1 was described, as the open state of Switch I and the closed state of Switch II, whereas the target state-2 was described as the partially open state of Switch I and the open state of Switch II. The target state-3 corresponded to the open state of Switch I and II as shown in *Figure 8*.

We applied the PSP method (*Jalalypour et al., 2020*) on the three scenarios given in *Figure 8*. The results showed that transition between the initial state and the target state-1 gave the highest overlap compared to other two states of the final state as shown in *Table 3*. Therefore, this finding shows that Switch I residues mainly contributed to the conformational transition of displacement of Switch I out of the nucleotide-binding pocket in HRAS$^{G12D}$.

We further investigated if cerubidine facilitated displacement of Switch I in terms of the energetic cost required. To this end, we performed steered molecular dynamics simulations by using ligand-free and cerubidine-bound HRAS$^{G12D}$ systems using the coordinates obtained by PRS method as shown in bold in *Table 3*. Y32 and its best direction with overlap values ($O^i$) of 0.72 were fed to SMD simulation. The initial structure was then perturbed by pulling the Cα atom of Y32 along this direction towards the target state-1. Each simulation was repeated 70 times and the potential of mean force (PMF) was calculated. We note that in the ligand-free simulations, 9 of the pullings led to diverging paths and were excluded from the analyses as they caused erroneous free energy287 estimates (see *Figure 9B*), *Bray et al., 2022*. Conversely, in the ligand-bound SMD simulations, all simulations were on the path (see *Figure 9B*). Results displayed in *Figure 9* indicate that while the ligand-free Ras favors the open form by 3.5 kcal/mol, there is a high barrier to the opening of the cavity of ca. 30 kcal/mol. In the presence of cerubidine, the opening is achieved with a barrierless transition. This finding shows that cerubidine facilitates the lowering of the potential barrier to the opening of Switch I and exposure of Y32.

## HEK-293T cells were engineered to express HRAS$^{G12D}$ mutant

To investigate G12D-specific system properties in vitro, we established G12D mutant HRAS expressing cell lines. Human embryonic kidney cells (HEK-293T; CRL-11268, ATCC) is a widely used cell line for gene delivery studies due to their high transfection efficiencies (*Ooi et al., 2016*). Accordingly, as proof of concept, we aimed to introduce G12D mutant HRAS into HEK-293T cells (293T-HRAS$^{G12D}$). We firstly subcloned the HRAS$^{G12D}$ gene region in the commercially available plasmid with a bacterial

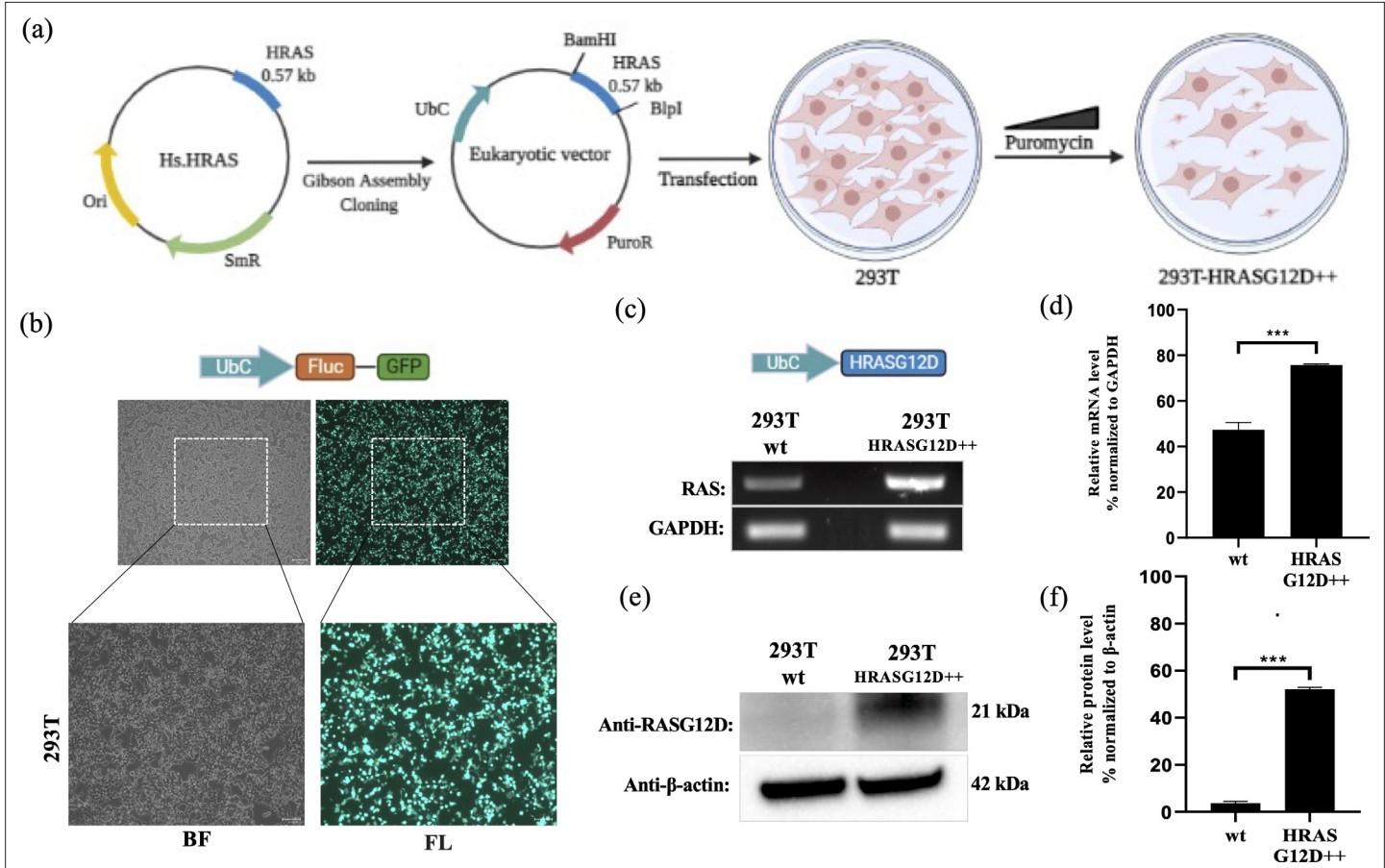

**Figure 10.** Engineering HEK-293T cells expressing mutant HRAS$^{G12D}$. (**a**) Schematic representation of the cloning HRAS$^{G12D}$ gene region into the eukaryotic expression plasmid (with PuroR gene to select transgene positive population) using the Gibson Assembly method and engineering HEK-293T cell line to overexpress HRAS$^{G12D}$ protein upon transfection followed by puromycin selection. (**b**) Fluorescent images 293T cells transfected with GFP-encoding plasmid. (**c**) RT-PCR analysis showing expression levels of HRAS$^{G12D}$ in 293T cells transfected with HRAS$^{G12D}$ plasmid. (**d**) ImageJ analysis of band densities from 'C'. (**e**) Western blot analysis showing expression levels of HRAS$^{G12D}$ in 293T-HRAS$^{G12D}$ cells. (**f**) ImageJ analysis of western-blot band densities. Data represent the means of three independent assays. Unpaired t-test analysis was used to test the difference between each experimental group and the control group. BF: bright field, FL: Fluorescence, ***: p<0,0001.

expression system into the eukaryotic expression plasmid carrying a puroR gene as a selection marker (See *Figure 10A*). Next, we showed that the transfection method reaches a high efficiency (90–95%) when a GFP expressing plasmid is introduced into HEK-293T cells (See *Figure 10B*).

A cDNA library of the 293T-HRAS$^{G12D}$ cell lysates was obtained for RT-PCR analysis and we detected that the 293T-HRAS$^{G12D}$ cells express increased levels of HRAS transcripts compared to control 293T cells (wild-type; cells with no gene transfer) (See *Figure 10C and D*). The primer sets can amplify both wild-type and mutant forms of HRAS since there is only one single base difference and no mutant specificity (*Muñoz-Maldonado et al., 2019*). Therefore, we detected HRAS$^{G12D}$ expression at the protein level using a G12D specific antibody. Our results showed that the transfected cells (293T-HRAS$^{G12D}$) express significantly high levels of HRAS$^{G12D}$ protein compared to wild-type cells (See *Figure 10E and F*). Interestingly, we found out that wild-type HEK-293T cells naturally lack G12D mutant protein expressions.

## Cerubidine treatment selectively inhibits the HRAS$^{G12D}$-RAF interaction and blocks activation of HRAS$^{G12D}$

To study the potential HRAS$^{G12D}$-RAF targeting effects of our proposed small molecule cerubidine, firstly, we determined the optimum doses of the molecule in 293T cell lines. The cells treated with a range of compound concentrations (1, 5, 10, 25, 50, 100 μM) showed 80% viability up to 10 μM

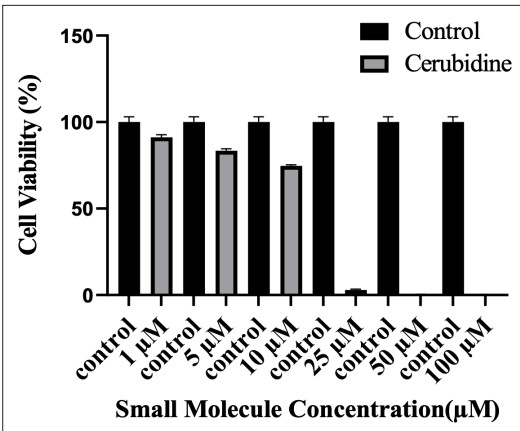

**Figure 11.** Cell viability assay in control and 24 hr treated 293T cells. The plot indicates the cell viability of 293T cells upon treatment with cerubidine in a dose-dependent manner (1, 5, 10, 25, 50, 100 μM).

treatment. Besides that, 25 μM and above cerubidine treatments were cytotoxic to the cells (See *Figure 11*). We then used active RAS pull-down and detection kit (Thermo) to analyze the interaction of the active RAS protein with RAF protein in the presence of cerubidine. To confirm the proper functioning of the kit, we treated 293T cell lysates with GTPγS and GDP in-vitro to activate and inactivate RAS. GTPγS is the non-hydrolyzable or slowly hydrolyzable analog of GTP. RAS is active when interacting with GTP and inactive upon binding of GDP (*Simanshu et al., 2017*). In this context, GTPγS was treated with RAS, which increased the interaction of the RAS protein with RAF by keeping it in its active form (See *Figure 12A,B and C*). Following detection of the RAS-RAF interaction, we treated 293T-HRAS^G12D cells with optimum doses of cerubidine and collected lysate for protein isolation at different time points. We detected a significant decrease in the active RAS$^{G12D}$, especially at the 12$^{th}$ hr of treatment. Additionally, we analyzed the presence/decrease of active wild-type HRAS in the same line and there was no significant change in active HRAS levels after cerubidine treatment. Overall data showed that the cerubidine treatment blocks HRAS-RAF interaction in a G12D-specific manner (See *Figure 12D, E and F*).

## Discussion

Due to involvement in crucial biological processes such as cell growth, proliferation, and differentiation, the RAS protein family has been used as a hot target in drug discovery studies. However, no therapeutic molecule has yet been proven to be used in the clinics due to the absence of deep clefts on the surface of the protein. On the other hand, recently, phosphorylation has been shown to impact the function of the RAS by inhibiting its interaction with effector proteins like RAF, which is involved in the onset of various cancer types. Moreover, examination of the crystal structures pertaining to RAS/RAF complexes showed that Y32 was pointing towards the nucleotide-binding pocket, whereas it was far in the RAF inhibitor-bound RAS protein (PDB ID: 6WGN) (*Zhang et al., 2020*) suggesting that orientational preference of Y32 might control interaction of RAS with RAF (See *Figure 13*).

In this study, motivated by these structural and biochemical data we set out to investigate the impact of phosphorylation on dynamics and structure of HRAS$^{WT}$, and aimed to induce similar modulation in the mutant HRAS by means of small therapeutics to prevent interaction with RAF. To this end, we performed extensive MD simulations on the phosphorylated HRAS and showed that the post-translational modification impacted the dynamics of Switch I and also pushed Y32 out of the nucleotide-binding pocket. Importantly, flexibility of Switch I in the mutant RAS provided a possible binding pocket in the vicinity of the nucleotide-binding site which could be targeted by FDA-approved ligands that modulated dynamics of Y32. Moreover, we also showed that displacement of Switch I and Y32 by the ligand was energetically more favorable than in the absence of the ligand.

Cancer cells show highly mutagenic profile and hard to treat with standard therapies without cancer cell selectivity. Additionally, in today's medicine, personalized approaches are ultimately needed considering the individual based differences of the pathology. HRAS mutations are very common in cancer and G12D variant is primarily found in bladder urothelial carcinoma, cutaneous melanoma, infiltrating renal pelvis, ureter urothelial carcinoma, melanoma, and colorectal adenocarcinoma (*The AACR Project GENIE Consortium et al., 2017*). Accordingly, in our study, we showed that cancer specific G12D mutant can be targeted by small molecules to interfere with RAF interaction and eventually RAS inactivation. Targeting HRAS-G12D by small molecules can be adopted to further study cell proliferation/death kinetics considering inhibition of RAF/MEK/ERK signalling. Here, we studied HRAS$^{G12D}$; however, high-sequence conservation and phosphorylation present among RAS isoforms suggest the potential application of the methodology to other members in the RAS protein family.

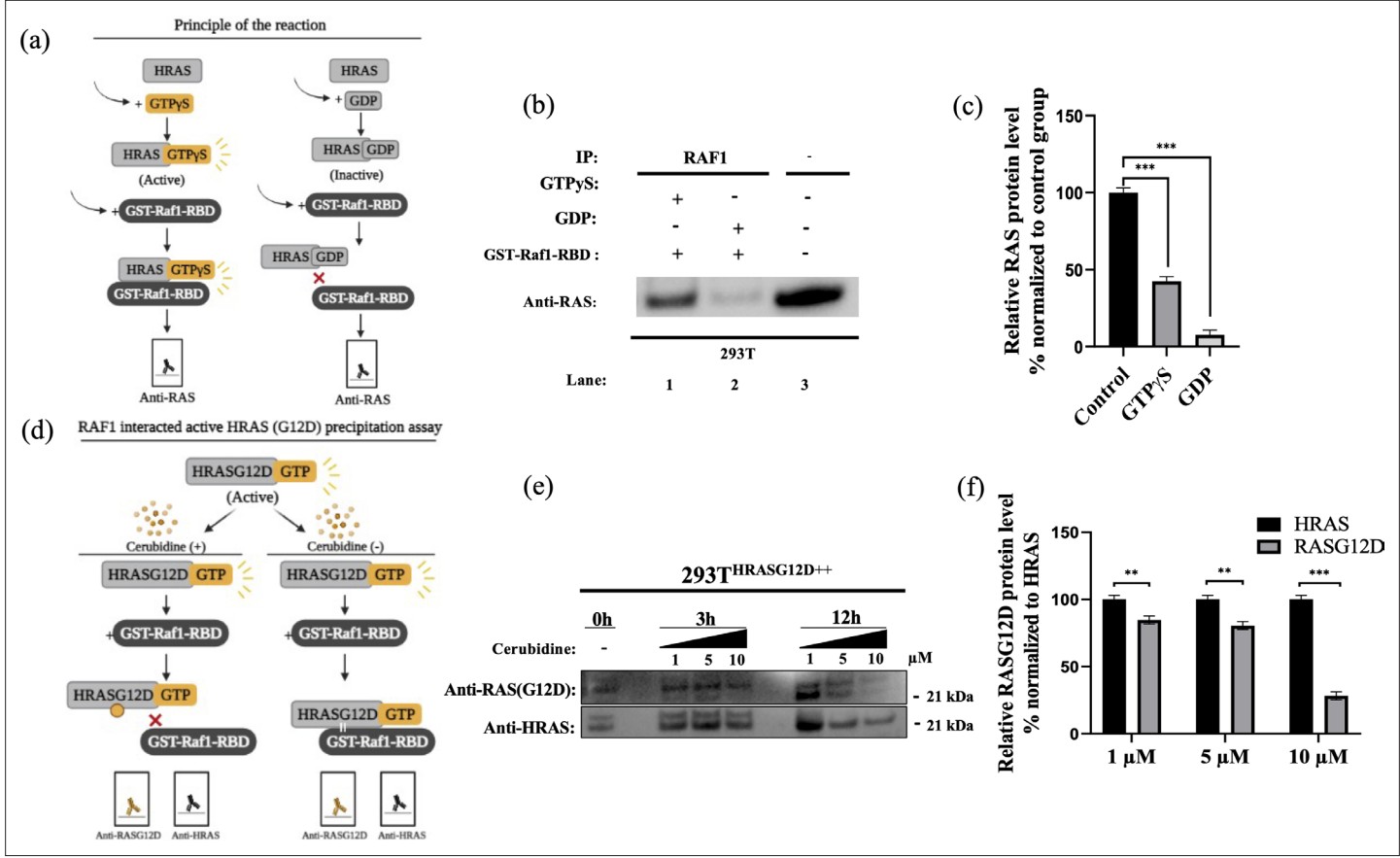

**Figure 12.** Cell viability assay in control and 24 hr treated 293T cells. (**a**) Scheme to outline the principle of active Ras pull-down reaction. (**b**) Immunoprecipitation (IP) assays show interactions of RAS with RAF proteins in the presence (Lane 1–2) and absence (Lane 3) of GTPγS-GDP. Protein extracts were immunoprecipitated with Raf1-RBD probe and resolved by SDS PAGE. Protein-protein interactions were immunodetected using anti-RAS antibodies. (**c**) ImageJ analysis of Western-blot band densities. Unpaired t-test analysis was used to test the difference between each experimental group and the control group. (**d**) Scheme outlining the RAF1 interacted active HRAS$^{G12D}$ precipitation assay. (**e**) Immunoprecipitation (IP) assays showing interactions of RAS with RAF proteins in 293T$^{HRASG12D++}$ cells treated with increasing doses (1,5 and 10 μM) of Cerubudine. Protein extracts obtained at different time points (0 hr, 3 hr, and 12 hr) were immunoprecipitated with the RAF1-RBD probe and resolved by SDS-PAGE. Protein-protein interactions were immunodetected using anti-RAS$^{G12D}$ and anti-HRAS antibodies (**f**) ImageJ analysis of Western-blot band densities. Unpaired t-test analysis was used to test the difference between each RAS$^{G12D}$ group and the HRAS group. **: p<0.001, ***: p<0.0001.

From that perspective, this study does not only provide mechanistic insight into the impact of phosphorylation but also opens up new avenues for possible use of the post-translational modification in future drug discovery studies. Hereby, we suggest further preclinical examination of our hypothesis for biological mechanisms which might potentiate their clinical uses.

# Materials and methods
## Molecular dynamics simulations of GTP-bound HRAS$^{pY32}$
### System setup for molecular dynamics simulations
The crystal structure of phosphoaminophosphonic acid-guanylate ester (GNP)-bound HRAS$^{WT}$ (PDB ID: 5P21) (*Pai et al., 1990*) was retrieved from the Protein Data Bank (https://www.rcsb.org/) (*Berman, 2000*; *Burley et al., 2019*). In order to prepare its GTP-bound state, the N$_3$B atom of GNP was substituted with oxygen atom. The crystal waters, which were located within 5 Å of the nucleotide, were kept in simulations. Following, the GTP-bound form of the protein was protonated at pH 7.4 according to the pKa values obtained from the ProPka server (*Søndergaard et al., 2011*; *Olsson et al., 2011*). The phosphorylation of Y32 residue was made using the TP2 patch provided by CHARMM-GUI server (*Johnson and Lewis, 2001*). The protein, GTP and Mg$^{2+}$ ion were parametrized

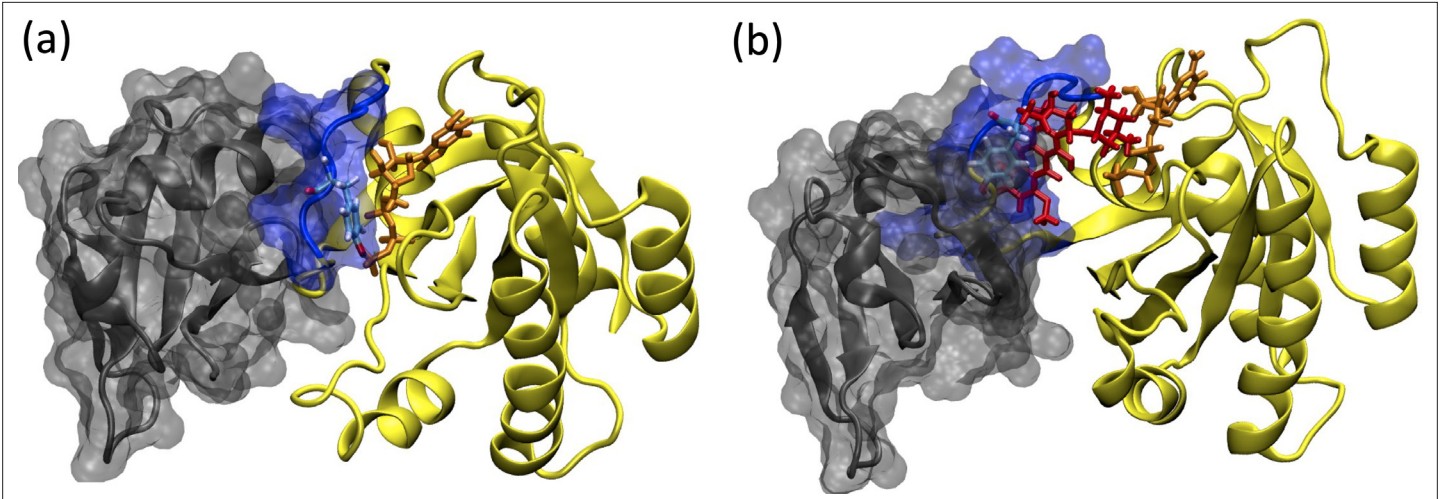

**Figure 13.** Depiction of the RAF/RAS interaction interface in the absence/presence of the ligand, cerubidine. (**a**) RAF-RBD in complex with HRAS. Y32 and GTP are shown in licorice representation, whereas protein and RAF-RBD interaction interface is shown in New Cartoon, and surface representation, respectively. (**b**) The displacement of Y32 from the nucleotide-binding pocket by cerubidine, which is colored with red, causes steric clash at the RAS/RAF interface.

using the CHARMM36m force-field (*Best et al., 2012*) while water molecules were modeled using the TIP3P water model (*Mark and Nilsson, 2001*). The thickness of the water layer was set to 15 Å to take periodic boundary conditions into account. Eventually, the solvated system was neutralized with 0.15 M NaCl.

## Simulation protocol

The MD simulations were employed via Compute Unified Device Architecture (CUDA) version of NAnoscale Molecular Dynamics (NAMD) (*Vanommeslaeghe et al., 2010*; *Best et al., 2012*; *Vanommeslaeghe and MacKerell, 2012*; *Vanommeslaeghe et al., 2012*; *Yu et al., 2012*; *Soteras Gutiérrez et al., 2016*), in which the graphical processing unit acceleration was enabled. Temperature, pressure, and time step were set to 310 K, 1 atm, 2 femtoseconds, respectively. In order to calculate the long-range electrostatic interactions, the particle mesh Ewald method was used (*Darden, 1993*; *Essmann et al., 1995*). For computation of non-bonded interactions, the cut-off value was adjusted to 12 Å. Moreover, the prepared system was minimized for 2400 time steps. After minimization, the GTP-bound HRAS$^{pY32}$ system was simulated in the NPT ensemble. Four simulations were performed per each system studied, each of which started with a different velocity distribution. We performed at least *ca.* 1 μs per each replicate of the system except some ligand-bound systems. For those, when the opening of Switch I was observed the simulation continued additional *ca.* 200 ns and then ended. Obtained trajectories were analyzed by combining these four replicates. Also, time-line data per each replicate is provided as the supplementary information (*Figure 2—figure supplement 1*, *Figure 3—figure supplement 1*, *Figure 4—figure supplement 2*, *Figure 5—figure supplement 1*, *Figure 7—figure supplement 1*, and *Figure 7—figure supplement 2*).

## Ensemble-based virtual screening

### Clustering the trajectory pertaining to HRAS$^{G12D}$, identification of possible binding pockets, and determination of pharmacophore groups

The most probable conformational state of the binding pocket pertaining to HRAS$^{G12D}$ was determined by using following reaction coordinates: distance measured between (i) side-chain oxygen of residue T35 and P$\gamma$ atom of GTP, (ii) backbone amide of the residue G60 and P$\gamma$ atom of GTP, which were used to point out different conformational states of the nucleotide-bindng pocket in our previous study (*Ilter and Sensoy, 2019*). The frames, which represent different conformational states of the nucleotide-binding pocket with respect to the above-mentioned coordinates, were selected. Subsequently, GTP and Mg$^{2+}$ were removed from the frames and proteins were optimized using the

OPLS3e force-field (*Roos et al., 2019*) that is available in the 'Protein Preparation' module of the Schrödinger software (*Sastry et al., 2013*; *Release, 2018*; *Roos et al., 2019*). The optimized structures were provided as inputs to the 'SiteMap' module of the Schrödinger (*Halgren, 2007*; *Halgren, 2009*; *Release, 2018*). Subsequently, possible binding pockets having higher scores were identified and utilized in further steps. Afterwards, pharmacophore groups were built up in accordance with chemical and geometrical properties of the identified binding pockets. To do so, the 'Develop Pharmacophore Model' module of Schrödinger was utilized (*Salam et al., 2009*; *Loving et al., 2009*). Following, candidate molecules, which include at least 3 of the 7 pharmacophore features and have molecular weight less than 550 kDa, were sought in the BindingDB (*Gilson et al., 2016*; *Liu et al., 2007*; *Chen et al., 2001b*; *Chen et al., 2001a*; *Chen et al., 2002*, DrugCentral *Ursu et al., 2019*; *Ursu et al., 2017*, NCGC *Huang et al., 2011*, and DrugBank *Wishart et al., 2018*; *Law et al., 2014*; *Knox et al., 2011*; *Wishart et al., 2008*; *Wishart et al., 2006* databases).

## Testing the stability of ligand-HRAS$^{G12D}$ complexes via atomistic simulations

After the selection of candidates based on their GScore values and orientations next to the nucleotide-binding pocket, the stability and the impact of the ligands on the structure and dynamics of HRAS$^{G12D}$ were explored by means of MD simulations. To this end, the topology and parameter files of the candidate molecules were prepared using the 'Ligand Reader & Modeler' of CHARMM-GUI (*Jo et al., 2008*; *Kim et al., 2017*). The systems were simulated using at least two replicates, each of which started with different initial velocity distribution under the same conditions that were used for HRAS$^{pY32}$.

## Local and global analysis of the trajectories

The trajectories were visualized with the 'Visual Molecular Dynamics' (VMD) and snapshots were rendered using the 'Taychon Render' (*Humphrey et al., 1996*; *Stone, 1998*). 'Groningen Machine for Chemical Simulations' (GROMACS) package and ProDy library were utilized for the local and global trajectory analysis (*Abraham et al., 2015*; *Lindahl and van der Spoel, 2021*; *Bakan et al., 2011*). Data were visualized using the Seaborn and Matplotlib libraries *Hunter, 2007*.

### Root-mean-square fluctuation

The root-mean-square fluctuation (RMSF) of backbone atoms throughout the obtained trajectories was calculated using the 'gmx rmsf' module of GROMACS (*Abraham et al., 2015*; *Lindahl and van der Spoel, 2021*) as shown in the below formula;

$$RMSF = \sqrt{(1/T)\sum_{t=1}^{N}(R_i(t) - \overline{R_t})} \qquad (1)$$

where $T$ and $R_i(t)$ correspond to the duration of simulation and coordinates of backbone atom $R_i$ at time $t$, respectively. By courtesy of this, the flexibility of each residue was computed, and made a holistic comparison with the systems. Particularly, for uncovering the impact of ligands on the backbone RMSF of Y32 and RAF interaction interfaces, the backbone RMSF value of the regions/residues of interest pertaining to ligand-bound HRAS$^{G12D}$ was subtracted from those of HRAS$^{G12D}$.

### probability distribution

The probability density distributions of atom-pair distances were exploited to have a closer look into the impact of the tyrosyl phosphorylation on HRAS$^{WT}$ as well as the impact of candidate molecules on HRAS$^{G12D}$. To this end, the 'gmx distance' module of GROMACS was utilized for measuring the distance (i) between the Cα atoms of G/D12 and P34, and (ii) between the side-chain oxygen atom of Y32—OH—and Pγ atom of GTP (*Abraham et al., 2015*; *Lindahl and van der Spoel, 2021*). The computed raw-data was converted into normalized probability density distribution plots by calculating the frequencies of the sampled distances adjusting the sampling range as 1Å via Matplotlib' library *Hunter, 2007*. As a side note, the sampling range of angle was set to 20°.

The probability density distribution of the distance between (i) the Oγ atom of T35 and Pγ of GTP, (ii) the backbone amide of G60 and Pγ of GTP, and (iii) Oε atom of Q61 and Pγ of GTP were also

drawn. In addition to the interatomic distances, the probability density distribution of $\chi_{22}$ pertaining to pY32 was also calculated.

## Number of water molecules

The number of water molecules around the GTP was calculated over the course of produced trajectories to reveal the impact of mutation, tyrosyl phosphorylation, and ligands on the exposure of GTP to the possible nucleophilic water attacks via the ProDy library (*Bakan et al., 2011*). To this end, the water molecules within 5 Å of GTP were selected and computed per frame. Thereafter, the mean of the number of water molecules around GTP was taken as well as the standard error of the mean was calculated.

## Principal component analysis

In addition to the local dynamics and structural properties of the phosphorylated system, its overall dynamics were also scrutinized via principal component analysis (PCA). The principal components of HRAS$^{pY32}$ were compared with those of HRAS$^{WT}$, and HRAS$^{G12D}$. By doing so, the collective effect of the tyrosyl phosphorylation was demystified. In this regard, the trajectory of HRAS$^{pY32}$ was aligned with respect to the Cα atoms of the reference structure, and subsequently, a diagonalized co-variance matrix was generated;

$$C_{jk} = \langle M_{jk}\Delta r_j \Delta r_k \rangle \tag{2}$$

where $M_{jk}\Delta r_j \Delta r_k$ corresponds to displacement from time-averaged structure for each coordinate of $j$ and $k$ atoms, whilst co-variance matrix is abbreviated by $C_{jk}$.

Following the generation of the diagonalized co-variance matrix, eigenvectors ($v$) and eigenvalues ($\delta^2$) were calculated.

$$C_{jk} = \delta^2 v \tag{3}$$

The diagonalized co-variance matrix was generated using the 'gmx covar' module of GROMACS (*Abraham et al., 2015*; *Lindahl and van der Spoel, 2021*). Thereafter, the 'gmx anaeig' module of GROMACS was made use of taking the projection of the trajectory with respect to the eigenvectors of interest, which eventually illuminated the collective spatial organization of the protein as well as the eigen RMSF values of the Cα atoms (*Abraham et al., 2015*; *Lindahl and van der Spoel, 2021*).

## Perturb-scan-pull

PSP consists of three parts, which are PRS, steered molecular dynamics (SMD), and potential of mean force (PMF) calculation (*Jalalypour et al., 2020*). Firstly, the PRS calculation of all the ligand-bound systems were conducted, whilst the SMD and PMF calculation were carried out for the cerubidine-bound HRAS$^{G12D}$ system, whose dynamic and structural properties are similar to those of other studied ligand-bound systems as elucidated by atomistic simulations.

### Perturbation-response scanning

PRS was performed to achieve the target states by perturbing each residues on the initial state, which, in turn, provided insight into the response of all residues in the HRAS$^{G12D}$. In this way, the residues, which play a pivotal role in the anticipated transitions, were aimed to be identified. To this end, the spatial position of both Switch I and II was clustered to determine initial and target states by measuring the distance between (i)the Cα atoms of D12 and P34 and (ii) the backbone amide of G60 and Pγ of GTP over the course of trajectories pertaining to HRAS$^{G12D}$ and ligand-bound HRAS$^{G12D}$. Following, the coarse-grain representation of each state was modeled by selecting the center of mass of the Cα atom pertaining to each residue as a node. Herein, 1000 random forces (ΔF) in distinct directions were sequentially exerted on each node in order to perturb the initial structure (*Atilgan and Atilgan, 2009*). In light of the linear response theory, displacement (ΔR) as a response to force exerted on the structure was derived from an equilibrated chunk of MD simulations;

$$\Delta \mathbf{R}_1 = \langle \mathbf{R} \rangle_1 - \langle \mathbf{R} \rangle_0 \cong \frac{1}{k_B T}\langle \Delta \mathbf{R}\Delta \mathbf{R}^T \rangle_0 \Delta \mathbf{F} = \frac{1}{k_B T}\mathbf{C}\Delta \mathbf{F} \tag{4}$$

where $R_0$ and $R_1$ correspond to the unperturbed initial state of HRAS$^{G12D}$ and perturbed predicted coordinates, respectively;

$$\mathbf{C} = \langle \Delta\mathbf{R}\Delta\mathbf{R}^T \rangle_0 \tag{5}$$

where the cross-correlation of the fluctuations of the nodes in the initial state is denoted by C.

$$O^i = \frac{\Delta\mathbf{R}^i \cdot \Delta\mathbf{S}}{|(\Delta\mathbf{R} \cdot \Delta\mathbf{R})^i(\Delta\mathbf{S} \cdot \Delta\mathbf{S})|^{1/2}} \tag{6}$$

The measured difference between the initial and target structures and the overlap between two nodes are denoted by $\Delta S$ and $O^i$, respectively.

## Steered molecular dynamics

Following the PRS calculation, SMD simulations were employed under the same circumstances as the above-mentioned MD simulations pertaining to HRAS$^{pY32}$. The set of external poses were imposed to the Cα atom of Y32, where the constant velocity and spring constant were adjusted to 0.03 Å ps$^{-1}$ and 90 kcal mol$^{-1}$Å$^{-2}$, respectively. Moreover, the Cα atoms of L23 and R149 residues were fixed along the pulling direction so as to prevent dislocation and rotation on the structure. The SMD runs were considered completed as long as the secondary structure of the protein was maintained and the final structure resembled the target conformation. With the time step of the simulations of 2 fs, this leads to covering a distance of ca. 40 Å in the pulling direction in ca. 3 ns. Data were recorded every 12 steps for further processing.

## Potential of mean force

The energy landscape of the transition in either presence or lack of the drug molecule, namely ceru-budine, was elaborated by calculating the PMF along the pulling direction. Considering the well-established procedure (*Jalalypour et al., 2020*), the PMF was computed according to the second-order cumulant expansion formula via,

$$F_{\lambda(t)} - F_{\lambda(0)} = \langle W(t) \rangle - \frac{1}{2k_BT}(\langle W(t)^2 \rangle - \langle W(t) \rangle^2) + \ldots \tag{7}$$

For either of the ceribudine-bound and ceribudine-free systems, 70 SMD simulations were conducted. The resulting trajectories were monitored so as to ensure they did not diverge from the intended path. In the ceribudine-bound systems, all trajectories stayed on the path, while the 9 simulations that diverged were discarded from the analyses in the ligand-free simulations. Curves were calculated by binning the displacement vs. force data, each projected on the pulling direction at intervals of 24 fs, into reaction coordinate distances of size 0.4 Å. Error bars were calculated by block averaging the data in each bin as well as bootstrapping (see *Figure 9—figure supplement 1*). SMD data and the programs used in their analyses are accessible at https://github.com/midstlab/PMF_from_SMD_data/, (copy archived at swh:1:rev:f72f6cbe253ddc4515b47f670eb936a50adc7824; *MIDST lab, 2022*).

## Preparation of plasmid constructs encoding HRAS$^{G12D}$

The bacterial expression plasmid Hs.HRAS$^{G12D}$ (83183) was obtained from Addgene (U.S). The mutant HRAS$^{G12D}$ was then inserted between the BamHI and BlpI restriction sites under the (UbC promoter) into the lentiviral vectors with the PuroR gene. (The vector was a kind gift from Dr. Shah (Brigham and Woman's Hospital, Harvard Medical School, Boston, U.S.) and was previously characterized and widely studied *Stuckey et al., 2015*).

## Engineering HRAS$^{G12D}$ expressing HEK-293T cells; 293T-HRAS$^{G12D}$

To investigate the in vitro outcomes of our in silico findings, HEK-293T cells (CRL-11268, ATCC) were engineered to express mutant HRAS$^{G12D}$. HEK-293T cell lines cultured on T75 flask with high-glucose DMEM medium which contains 10% fetal bovine serum (FBS), 1% Penicillin Streptomycin at 37 °C in 5% CO$_2$ incubator. One day (18–24 hr) prior to transfection, cells were seeded at an optimum density that reaches 70–80% confluency the next day, at the time of transfection. Plasmid DNA was transfected into cells using Trans-Hi In Vitro DNA Transfection Reagent (F90101TH, FormuMax) according to the

manufacturer's recommendations. 12–18 hr after transfection, the medium containing the Trans-Hi/DNA complex was removed and replaced with a fresh whole serum/antibiotic-containing medium.

## HRAS[G12D] expression analysis

To use 293T-HRAS[G12D] cells in the following experiments, first of all, we analyzed the overexpression of HRAS transcripts by reverse transcriptase-polymerase chain reaction(RT-PCR). The primer sets can not be specific to the mutant G12D since there is only 1 single base difference in G12D mutant versus wild-type HRAS. For this reason, we further evaluated G12D expression at the protein level via western blot.

### RT-PCR

To verify increased HRAS transcript levels in HEK-293T cells by RT-PCR, we firstly harvested cells expressing HRAS[WT] and HRAS[G12D] to prepare RNA samples. Afterward, RNA was extracted using RNeasy Mini Kit (74104, Qiagen) and the cDNA library was prepared from 1 µg of total RNA, using SuperScript VILO cDNA Synthesis kit (11754050, Invitrogen). HRAS was then amplified by RT-PCR using a standard PCR protocol on a T100 Thermal Cycler (BIO-RAD). Gene expression was normalized to that of a housekeeping gene; GAPDH (glyceraldehyde-3-phosphate dehydrogenase). The primer sets (5'–3') used for RT-PCR were as follows:

*GAPDH: Fw- GTCAGTGGTGGACCTGACCT; Rv- TGCTGTAGCCAAATTCGTTG* (245 bp PCR product) and *HRAS: Fw- GGATCCATGACGGAATATAAGCTGG; Rv- GCTCAGCTTAGGAGAGCACACACTTGC* (570 bp PCR product).

### Protein sample preparation

Cells were washed two times with ice-cold phosphate-buffered saline (PBS) prior to 1 X lysis buffer (25 mM Tris-HCl,150 mM NaCl, 5 mM $MgCl_2$,1%NP-40, and 5% glycerol) involving complete Mini Protease Inhibitor Cocktail tablet (11836153001, Roche). Lysates were spun at 16,000×g for 15 min and the supernatants were reserved as protein samples. The Pierce BCA Protein Assay Reagent (23227, Thermo Fisher Scientific) was used to quantify the protein concentration of each sample.

### Western blot (WB) analysis

Cell lysates were resolved by sodium dodecyl sulfate (SDS)-polyacrylamide gel electrophoresis (PAGE) using Bolt 4 to 12%, Bis-Tris, 1.0 mm, Mini Protein gel (NW04120BOX, Invitrogen). Proteins were transferred into nitrocellulose membrane by iBlot 2 Dry Blotting System (Invitrogen) at constant current of 1.3 A for 7 min. Membranes were blocked with 5% BSA-ALBUMIN in Tris-buffered saline/0.1% Tween-20 for 1 hr at room temperature and incubated overnight with rabbit anti-RAS[G12D] (mutant specific) (14429, Cell Signaling Technology) antibody. After primary antibody incubation, membranes were washed with TBST (Tris-buffered saline with 0.05% Tween-20). Secondary antibody (R-05071–500, Advansta), HRP conjugated goat anti-rabbit was diluted to 1:3000 in 5% BSA and incubated for 1 hr at room temperature. Membranes were developed using ECL substrate (1705061, Bio-Rad) and a chemiluminescence signal was detected by Chemidoc (Bio-Rad). Next, β-actin levels were determined as loading controls. For this, the membrane was incubated in stripping buffer (0.2 M Glycine, 0.10% Tween-20, pH:2.5) and blocking solution before reprobing with anti-β-actin (3700, Cell Signaling Technology).

## Optimization of optimum cerubidine treatment doses to target HRAS[G12D]-RAF interaction

HEK-293T cells were plated at 5000 cells/well into 96 black well plates (3603, Corning) and cultured in DMEM, high glucose (Gibco) containing 10% FBS at 37 °C in 5% CO2. Cells were cultured overnight and the compounds (dissolved in DMSO) were added to the cells at concentrations ranging from 0 to 100 µM. The cells were incubated under standard culture conditions for 24 hr. Cell viability was quantified using the CellTiterGlo Luminescent Cell Viability Assay (Promega) according to the manufacturer's instructions to measure ATP generated by metabolically active cells. Luminescent signals were measured using the SpectraMAX (Molecular Devices). The luminescence signals obtained from the compound-treated cells were normalized against the signal for DMSO-only treated cells.

## Active RAS pull-down assay

In this experimental setup, we conceptually investigated "G12D versus wild-type" HRAS presence in the active RAS population in the cells treated with cerubidine. RAS activity was determined using Active RAS Pull-Down and Detection Kit (Thermo Fisher Scientific) following the manufacturer's instructions. Firstly, we tested the assay validity using provided supplements. Lysates were incubated with gluta-thione S-transferase fusion of the RAS binding domain (RBD) of RAF1 along with glutathione agarose for 1 hr. Agarose beads were collected by centrifugation and washed three times with 1 X Wash Buffer (25 mM Tris-HCl,150 mM NaCl, 5 mM $MgCl_2$,1%NP-40, and 5%glycerol). Each sample was resuspended and boiled at 100 °C for 5 min. Samples were analyzed by western blotting as previously described. Analysis of RBD pull-down lysates was performed with mouse anti-RAS Antibody (16117, Thermo Fisher Scientific). Secondly, we prepared cell lysates from cerubidine treated 293T-HRAS$^{G12D}$ cells. One day prior to treatment plated a sufficient number of cells so that the cell density reaches the optimal conflu-ency (60–70%) at the time of treatment. Cells were incubated with increased cerubidine concentrations (1, 5, and 10 µM) for 3 hr and 12 hr (0 hr was used as control). After incubation, the active Ras pull-down assay was performed with proteins isolated from the treated and untreated cells (as described above in the protein sample preparation section). Finally, samples were subjected to western blotting as previously described. RBD pull-down lysates were probed with mouse anti-HRAS (sc-29, Santa Cruz Biotechnology), and rabbit anti-RAS$^{G12D}$ Mutant Specific antibodies (14429, Cell Signaling Technology).

## Acknowledgements

MI and OS thank TUBITAK and TUSEB for providing funding in the scope of 2209 A Undergrad-uate Research Support Program, reference number: 1919B011701434, and Computational Structural Biology Strategic R&D Project Call, reference number: 2019-TA-02–3561, respectively. CA thanks for partial support from TUBITAK, project no. 116F229. MI and OS also acknowledge Istanbul Medipol University for providing access to the High-Performance Computing System so as to run some of the MD simulations. MI, OS, FJ, and CA thank TUBITAK ULAKBIM for giving access to High Performance and Grid Computing Center (TRUBA resources) to complete the rest of the numerical calculations reported in this paper. NK thanks to Dr. Khalid Shah from Brigham and Woman's Hospital, Harvard Medical School, U.S. for the plasmid backbones.

## Additional information

### Funding

| Funder | Grant reference number | Author |
|---|---|---|
| Health Institute of Turkey | 3561/2019-TA-02 | Metehan Ilter<br>Ramazan Kasmer<br>Ozan Topcu<br>Nihal Karakas<br>Ozge Sensoy |
| The Scientific and Technological Research Council of Turkey | 116F229 | Farzaneh Jalalypour<br>Canan Atilgan |

The funders had no role in study design, data collection and interpretation, or the decision to submit the work for publication.

### Author contributions

Metehan Ilter, Conceptualization, Data curation, Formal analysis, Investigation, Methodology, Writing – original draft, Writing – review and editing; Ramazan Kasmer, Farzaneh Jalalypour, Conceptualiza-tion, Data curation, Formal analysis, Investigation, Methodology, Writing – review and editing; Canan Atilgan, Nihal Karakas, Conceptualization, Supervision, Investigation, Methodology, Writing – orig-inal draft, Writing – review and editing; Ozan Topcu, Methodology; Ozge Sensoy, Conceptualiza-tion, Supervision, Funding acquisition, Investigation, Writing – original draft, Project administration, Writing – review and editing

## Author ORCIDs
Canan Atilgan http://orcid.org/0000-0003-0557-6044
Ozge Sensoy http://orcid.org/0000-0001-5950-3436

### Decision letter and Author response
Decision letter https://doi.org/10.7554/eLife.79747.sa1
Author response https://doi.org/10.7554/eLife.79747.sa2

## Additional files

### Supplementary files
• MDAR checklist

• Supplementary file 1. Result of the analyses for the number of water molecules, PRS calculations and total simulation time for ligand-mutant HRAS trajectories. a The average number of water molecules within 5 Å of GTP were calculated over the course of the ligand-bound HRAS$^{G12D}$ systems. b Total simulation time performed for ligand- HRAS$^{G12D}$ complexes and changes in the backbone RMSF profiles of cerubidine-, tranilast-, nilotinib-, and epirubicin-bound HRAS$^{G12D}$ systems with respect to those of HRAS$^{G12D}$. c The results of PRS calculations for the transition between initial and target states.

• Source data 1. Raw unedited gel.

• Source data 2. Uncropped gels with relevant bands that are clearly labelled.

### Data availability
Simulated data used to generate the figures in the commentary are available at https://doi.org/10.17605/OSF.IO/Z2Y5S.

The following dataset was generated:

| Author(s) | Year | Dataset title | Dataset URL | Database and Identifier |
|---|---|---|---|---|
| Ilter M, Kasmer R, Jalalypour F, Atilgan C, Topcu O, Karakas N, Sensoy O | 2022 | Inhibition of mutant RAS-RAF interaction by mimicking structural and dynamic properties of phosphorylated RAS | https://doi.org/10.17605/OSF.IO/Z2Y5S | Open Science Framework, 10.17605/OSF.IO/Z2Y5S |

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
