## [Editor Report]

This study employs extensive MD simulations to probe the effect of phosphorylation of a tyrosine residue on the conformational ensemble of Ras GTPase. The insights form the basis for a screen of small molecule(s) that disrupt interaction with its target Raf kinase, and predictions are tested experimentally. Overall, the integrated approach is of interest to a wide range of biochemists and protein scientists and could potentially be used to modulate the activities of other proteins.

---

## [Decision Letter]

**Decision letter after peer review:**

Thank you for submitting your article "Inhibition of mutant RAS-RAF interaction by mimicking structural and dynamic properties of phosphorylated RAS" for consideration by *eLife*. Your article has been reviewed by 3 peer reviewers, one of whom is a member of our Board of Reviewing Editors, and the evaluation has been overseen by a Reviewing Editor and Jonathan Cooper as the Senior Editor. The following individual involved in the review of your submission has agreed to reveal their identity: Yuji Sugita (Reviewer #2).

Essential revisions:

The most essential revisions that the reviewers have suggested concern statistical error analysis of the simulation results, and correction of the computed free energy profile in Figure 9. Essentially all reviewers expressed major concerns about the correctness of the computed PMF, in terms of both the qualitative trend and quantitative values.

*Reviewer #1 (Recommendations for the authors):*

Overall, I think the simulations have been conducted properly and analyzed carefully. However, I have two major questions.

1. Foremost, the results in Figure 9 are potentially very confusing. The free energy scale here is several hundreds of kcal/mol, which is clearly not physical for the conformational rearrangement of the SwI motif. Second, even at a qualitative level, the PMF is downhill in nature without any obvious barrier. This is also largely unexpected. Finally, the downhill nature is less clear with the bound ligand, which doesn't readily explain how the ligand facilitates the conformational rearrangement in SwI as the authors claim. I think the PSP method is interesting but needs to be explained and discussed much more carefully, in terms of both qualitative features and quantitative values of the results.

2. On Pg6, the authors commented on the level of hydration of the γ phosphate in various protein variants, and suggested that a higher level of hydration can explain the higher level of intrinsic GTPase activity. This correlation is not immediately obvious to me, as the enzymatic activity is often inversely correlated with the level of active site hydration since a higher hydration level likely leads to higher reorganization energy and therefore a higher activation free energy barrier.

*Reviewer #2 (Recommendations for the authors):*

As I pointed out in the public review, I recommended the authors repeat the simulations a few times for improving the statistical significance and adding the errors. Also, I have several requests about the data presentations.

(1) In the distributions they showed in Figures (such as Figures 4C, 4D, 5A, 5B, and 5C), the curves are very smooth, lacking any noise. It seems to be unusual for me in MD simulations. I recommend the authors show the original data without smoothing, which is more natural for me (and readers).

(2) Clarification of structural figures is required. For instance, we can't see the green characters embedded in Figures 2E and 3B. Although the key residues are annotated in Figure 1A, I suggest the authors annotate key residues in other structural figures as well.

(3) PMF should be computed with enhanced sampling, such as umbrella sampling or metadynamics.

(4) More technical details are required in the Methods. Program package of MD simulation (Gromacs?) should be mentioned. I don't know which force field (AMBER of CHARMM, and which version) is used in the simulation.

*Reviewer #3 (Recommendations for the authors):*

The authors need to address the issues given in the public review by:

• Increasing their data range (at least five independent simulations in total) and/or proving convincingly that the data they collected is robust and the observed structural and dynamical effects are real. The most convincing way to do so is to include ensemble means and respective error bars in all distribution plots.

• Clearly stating which system (WT; pY32; G12D) has been simulated and how often for how much time.

• Showing data on the convergence of their results, e.g., RMSD plots of the protein backbone over time for the simulations.

• Either remove the PMF data completely or search for the pathways present in the data set (see the given article Jäger et al., J. Chem. Mol. Model. 2022) and analyze trajectories according to pathways. Additionally: how many independent pulling simulations did the authors perform? How long were these simulations?

[Editors' note: further revisions were suggested prior to acceptance, as described below.]

Thank you for resubmitting your work entitled "Inhibition of mutant RAS-RAF interaction by mimicking structural and dynamic properties of phosphorylated RAS" for further consideration by *eLife*. Your revised article has been evaluated by Jonathan Cooper (Senior Editor) and a Reviewing Editor.

The manuscript has been substantially improved but there are some remaining issues that need to be addressed. In particular, as Reviewer #3 highlighted, it is important to clarify several technical details, especially concerning the PMF calculations.

*Reviewer #1 (Recommendations for the authors):*

The authors significantly expanded the scope of the sampling in their simulations, especially the steered MD simulations. The results are more sound and generally support the mechanistic picture that emerged from the study of the effect of phosphorylation. I think the revision is appropriate for publication.

*Reviewer #2 (Recommendations for the authors):*

In the revised manuscript, the authors addressed almost all the issues that I had been concerned about in the original one. In particular, a very large free-energy barrier in figure 9 was a big problem in the original manuscript. To resolve this, the authors added more "pulling simulations" to get converged the PMF as shown in the revised manuscript. Although this is a different approach than I suggested, the results are now reasonable and fine for me. I expect that this has involved their significant computations in additional molecular dynamics simulations and I appreciate their effort in the revisions.

I also suggested the authors to improve the methods section for telling the technical points more clearly. According to the modified sentences (highlighted in yellow), it looks now fine.

Overall, the revised manuscript has been improved greatly compared to the original submission. The authors satisfactorily addressed the questions and comments by reviewers.

*Reviewer #3 (Recommendations for the authors):*

While the authors have addressed some of my concerns, not all points have been sufficiently addressed and need additional revisions:

• The distributions in Figures 2a,b; 3a; 4c,d; 5; 7 and most of the histograms in the SI still are missing error bars. Please do the following: Calculate distributions of distances or angles for each simulation separately. Then calculate the mean distributions as a function of distance/angle, as well as the respective standard error of the mean. Display the mean distributions as lines, and the standard error of the mean as a shaded area.

• Steered pulling: The PMF in Figure 9 looks much better now – too good for my experience, actually, as only 70 simulations were used as input for a 2nd cumulant expansion of the Jarzynski identity. And especially the error bars are incredibly small. I do not understand how the block averaging procedure mentioned in the text should work here. Usually, error estimation with the 2nd cumulant approximation is performed based on Jackknifing or bootstrapping – what do the respective errors look like when using this approach? What are the pulling rates employed? How was the "pathway analysis" performed? On what criteria were trajectories kept or discarded? And: The insight that different paths cause erroneous free energy estimates is not "well known", as is currently described in the text, but rather new insight – please cite, e.g., Bray et al., J. Chem. Inf. Model. 2022, 62, 4591-4604. In general, I agree here with reviewer #2 who requested Umbrella Sampling calculations instead of the steered MD results given here.

• PCA: I do not find the answer to my question why only the first three eigenvectors were used convincingly. If the authors want to leave out the remaining eigenvectors that contribute the remaining 30% of cumulative eigenvalues, they need to provide evidence of why this is applicable. They could, e.g., plot free energies calculated from the simulation data along the eigenvectors and show for which eigenvectors the distributions are non-trivial, i.e., not following a normal distribution.

---

## [Author Response]

Reviewer #1 (Recommendations for the authors):Overall, I think the simulations have been conducted properly and analyzed carefully. However, I have two major questions.1. Foremost, the results in Figure 9 are potentially very confusing. The free energy scale here is several hundreds of kcal/mol, which is clearly not physical for the conformational rearrangement of the SwI motif. Second, even at a qualitative level, the PMF is downhill in nature without any obvious barrier. This is also largely unexpected. Finally, the downhill nature is less clear with the bound ligand, which doesn't readily explain how the ligand facilitates the conformational rearrangement in SwI as the authors claim. I think the PSP method is interesting but needs to be explained and discussed much more carefully, in terms of both qualitative features and quantitative values of the results.

We thank the reviewer for this comment. We have now increased the number of SMD simulations to 70 (from the 10 in the original submission) for each system. The pulling directions remain the same. We find that in the ligand-free form of Ras, 9 of these trajectories diverge from the intended path and are discarded from the analyses. In the ligand-bound form, all trajectories stay on-path. We can now clearly show two features of the ligand-free Ras: The SwI open form is ca. 3.5 kcal/mol lower in energy; however, there is a ca. 30 kcal/mol barrier to the transition in this case. In cerubidine-bound Ras, there is no barrier to opening the switch and the energy required to reach this state is ca. 10 kcal/mol. We note that the open form is no longer the lower energy state due to the presence of the nearby cerubidine.

2. On Pg6, the authors commented on the level of hydration of the γ phosphate in various protein variants, and suggested that a higher level of hydration can explain the higher level of intrinsic GTPase activity. This correlation is not immediately obvious to me, as the enzymatic activity is often inversely correlated with the level of active site hydration since a higher hydration level likely leads to higher reorganization energy and therefore a higher activation free energy barrier.

We would like to thank the reviewer for pointing this out. It has been shown that anchoring Y32 to the _γ_-phosphate contributes to the capturing of the catalytic waters that participate in intrinsic hydrolysis (Structural impact of GTP binding on downstream KRAS signaling; Chem. Sci., 2020, 11, 9272-9289) by mediating a triple proton transfer step from the catalytic water-assisting water-Y32 to _γ_-phosphate. In the same study, the authors also showed that partial displacement of Q61 leads to opening-up of the catalytic pocket and destabilization of catalytic waters, thus impairing the intrinsic hydrolysis. Therefore, we agree with the reviewer that a higher level of hydration does not correlate with increased intrinsic GTPase activity of RAS. Bunda *et al.,* on the other hand, showed that phosphorylation of Y32 inhibits RAF binding, and promotes association of RAS with GAP hence promoting GTP hydrolysis. We showed that Switch I and II undergo conformational rearrangements upon phosphorylation. Considering the fact that GAP binding involves interaction with Switch I and II of RAS, it can be said that phosphorylation helps rearrangement of the nucleotide-binding pocket of RAS, hence modulating GTPase activity of the protein by mediating its interaction with GAP. The following sentence has therefore been removed in the revised version:

Line 145-146 of the original MS: thus, presumably, modulating intrinsic GTPase activity of the protein, and revised as follows:

“Considering rearrangements occurring around the nucleotide-binding pocket and that GAP binding involves interaction with both Switches I and II, it can be said that phosphorylation helps rearrangement of the nucleotide-binding pocket of RAS, hence modulating the GTPase activity of the protein by mediating its interaction with GAP” (Page: 5; Lines:150-154).

Reviewer #2 (Recommendations for the authors):As I pointed out in the public review, I recommended the authors repeat the simulations a few times for improving the statistical significance and adding the errors. Also, I have several requests about the data presentations.

We thank the reviewer for his suggestion, and we apologize for the lack of statistical significance information in the original submission. We repeated the simulations four times for each system studied and provided respective timelines as figure supplements in the revised version. We provided the standard errors in the figure captions.

(1) In the distributions they showed in Figures (such as Figures 4C, 4D, 5A, 5B, and 5C), the curves are very smooth, lacking any noise. It seems to be unusual for me in MD simulations. I recommend the authors show the original data without smoothing, which is more natural for me (and readers).

As the reviewer suggested, we provided the original data pertaining to Figures 4C, 4D, 5A, 5B, and 5C in Figure 4—figure supplement 2 & 3, and Figure 5—figure supplement 1 of the revised version and keep the smoothed ones in the main text.

(2) Clarification of structural figures is required. For instance, we can't see the green characters embedded in Figures 2E and 3B. Although the key residues are annotated in Figure 1A, I suggest the authors annotate key residues in other structural figures as well.

We have now annotated key residues in Figures 2C, 2D, 2E, and 3B in the revised version. Also, we have clarified the characters mentioned by the reviewer in revised Figures 2E and 3B.

(3) PMF should be computed with enhanced sampling, such as umbrella sampling or metadynamics.

Considering the options pointed out by all three reviewers regarding the sampling of the states to calculate the PMF, we chose to increase the number of samples from 10 to 70 for each system. The pulling directions remain the same. We find that in the ligand-free form of Ras, 9 of these trajectories diverge from the intended path and are discarded from the analyses. In the ligand-bound form, all trajectories stay on-path. We can now clearly show two features of the ligand-free Ras: The SwI open form is ca. 3.5 kcal/mol lower in energy; however, there is a ca. 30 kcal/mol barrier to the transition in this case. In cerubidine-bound Ras, there is no barrier to opening the switch and the energy required to reach this state is ca. 10 kcal/mol. We note that the open form is no longer the lower energy state due to the presence of the nearby cerubidine.

(4) More technical details are required in the Methods. Program package of MD simulation (Gromacs?) should be mentioned. I don't know which force field (AMBER of CHARMM, and which version) is used in the simulation.

We would like to thank the reviewer to point out the ambiguity regarding technical details. We mentioned the full name of the software package in the original text. Yet, the abbreviation of the software was not written therein. To clarify the technical details, we added the abbreviation of the simulation package along with its version in the revised version of the main text —the CUDA version of NAMD. We would like to point out however that the force fields used to model protein and water were given in the original manuscript. We used CHARMM36m and TIP3P to model protein/nucleotide and water, respectively (page 15 in the current version).

Reviewer #3 (Recommendations for the authors):The authors need to address the issues given in the public review by:• Increasing their data range (at least five independent simulations in total) and/or proving convincingly that the data they collected is robust and the observed structural and dynamical effects are real. The most convincing way to do so is to include ensemble means and respective error bars in all distribution plots.

During the revision period, we performed additional simulations to have four replicates, each of which was about 1 _µ_s, per system. For ligand-bound RAS systems, we ran the simulations until Switch I was displaced from the nucleotide-binding pocket and extended it for an additional *ca*. 200-300 ns to check if it comes back to its original position. Respective timeline plots of the replicates of both ligand-bound and non-liganded systems were provided as the figure supplements in the revised manuscript. We also reported error values in the caption of corresponding figures in the main text. Information regarding simulation times was provided in the methods section. Also, we revised the total simulation times for each ligand-bound RAS system in the SI of the revised manuscript.

Importantly, we observed similar behavior in each replicate of the systems; we, therefore, conclude that the results presented in the original manuscript are reproducible.

• Clearly stating which system (WT; pY32; G12D) has been simulated and how often for how much time.

As explained above, we performed 4 replicates of simulations for HRAS^WT^, HRAS^G12D,^ and HRAS^pY32^, each of which is about 1µs. Corresponding information was provided in detail in the methods section of the revised manuscript.

• Showing data on the convergence of their results, e.g., RMSD plots of the protein backbone over time for the simulations.

The backbone RMSD of the studied systems were provided in the response letter (See Figure 1 (vide supra)).

• Either remove the PMF data completely or search for the pathways present in the data set (see the given article Jäger et al., J. Chem. Mol. Model. 2022) and analyze trajectories according to pathways. Additionally: how many independent pulling simulations did the authors perform? How long were these simulations?

We have increased the number of simulations in the data set from 10 to 70 and we have analyzed the trajectories according to the pathways as suggested by the referee. The information on the length of the simulations, data recording and processing are now provided in more detail on page 18.

[Editors' note: further revisions were suggested prior to acceptance, as described below.]

The manuscript has been substantially improved but there are some remaining issues that need to be addressed. In particular, as Reviewer #3 highlighted, it is important to clarify several technical details, especially concerning the PMF calculations.Reviewer #3 (Recommendations for the authors):While the authors have addressed some of my concerns, not all points have been sufficiently addressed and need additional revisions:• The distributions in Figures 2a,b; 3a; 4c,d; 5; 7 and most of the histograms in the SI still are missing error bars. Please do the following: Calculate distributions of distances or angles for each simulation separately. Then calculate the mean distributions as a function of distance/angle, as well as the respective standard error of the mean. Display the mean distributions as lines, and the standard error of the mean as a shaded area.

We revised the corresponding histograms in the way the reviewer explained and presented them in the new version of the main text and the supplementary file.

• Steered pulling: The PMF in Figure 9 looks much better now – too good for my experience, actually, as only 70 simulations were used as input for a 2nd cumulant expansion of the Jarzynski identity. And especially the error bars are incredibly small. I do not understand how the block averaging procedure mentioned in the text should work here. Usually, error estimation with the 2nd cumulant approximation is performed based on Jackknifing or bootstrapping – what do the respective errors look like when using this approach? What are the pulling rates employed? How was the "pathway analysis" performed? On what criteria were trajectories kept or discarded? And: The insight that different paths cause erroneous free energy estimates is not "well known", as is currently described in the text, but rather new insight – please cite, e.g., Bray et al., J. Chem. Inf. Model. 2022, 62, 4591-4604. In general, I agree here with reviewer #2 who requested Umbrella Sampling calculations instead of the steered MD results given here.

In fact, the block averaging method was referenced in the first SMD paper to be used for error calculation: https://aip.scitation.org/doi/10.1063/1.1590311. As per reviewer`s request, we also estimated the error via bootstrapping method using 500 iterations, retaining 80% of the data in each calculation. The obtained profile is provided in Author response image 1. We also referenced this paper mentioned in the revised version and included the PMF profile along with the errors calculated by bootstrapping.

**Author response image 1. sa2fig1:** PMF profiles obtained by SMD experiments. The errors are calculated by both block averaging (left) and bootstrapping (right).

What are the pulling rates employed?

This information was already provided in the manuscript (page 18, line 504-512);

The set of external poses were imposed to the Cαα atom of Y32, where the constant velocity and spring constant were adjusted to 0.03 Å ps^-1^ and 90 kcal mol^-Å-2,^ respectively. Moreover, the Cαα atoms of L23 and R149 residues were fixed along the pulling direction so as to prevent dislocation and rotation on the structure.

How was the "pathway analysis" performed? On what criteria were trajectories kept or discarded?

Herein, we present Author response image 2 to show the pathway that the pulled atom followed in each case. The off-pathway trajectories can be directly seen from these figures; in particular, they all followed the same pathway with a narrow range when the ligand is bound. In the unliganded one, we now discard 7 trajectories, for the seven that stay away from the bundle. The two green dots that stay with the bundle are now included in the PMF calculation, with no difference in the final PMF profile.

**Author response image 2. sa2fig2:** The pathway analysis of SMD simulations. The displacement of the pulled atom, which is shown in green color, is shown throughout the 70 SMD trajectories in both ligand and ligand-free states. The protein is shown in new cartoon representation and colored according to the respective secondary structures. The trajectories excluded are shown in red while those included in green. The protein is aligned using all the regions except the loop.

In general, I agree here with reviewer #2 who requested Umbrella Sampling calculations instead of the steered MD results given here.

The pulling experiments support our claim and are not included to make a quantitative assessment of the observations. We therefore chose not to do the umbrella sampling calculations. Reviewer 2 is now satisfied with the findings; as they state, these have indeed involved significant computational resources.

• PCA: I do not find the answer to my question why only the first three eigenvectors were used convincingly. If the authors want to leave out the remaining eigenvectors that contribute the remaining 30% of cumulative eigenvalues, they need to provide evidence of why this is applicable. They could, e.g., plot free energies calculated from the simulation data along the eigenvectors and show for which eigenvectors the distributions are non-trivial, i.e., not following a normal distribution.

As a correction, we had presented first five eigenvectors in the revised version of the manuscript. As per reviewer`s request, we, hereby, present the first 28 eigenvectors that contribute to the 80% of the overall dynamics. As can be seen in Author response image 3, we observed peaks in the same regions, which were captured by the first five eigenvectors, using less dominant eigenvectors. Having this said, it is also important to mention we observed additional peaks for the region that was spanned by residues 100-150 yet with lower contribution to the dynamics. The hypothesis we proposed in the manuscript was to check if it was possible to prevent interaction between RAS and RAF by distorting the Switch I region. By means of PRS calculations, we showed that the optimum pathway to distort it would be the distortion of the region itself as it gave higher overlapping coefficient. The region spanned by residues 100-150 did not emerge as the one that could be used for a target site. Moreover, the possibility that a conformational change can be described by a few normal modes was also shown in the study of Petrone and Pande (Petrone P, Pande VS. Can conformational change be described by only a few normal modes? Biophysical journal. 2006; 90(5):1583–1593), whereby 50% cutoff was used to do all the relevant discussions. We therefore did not further modify this part in the manuscript. This paper is also referenced in the current version of the manuscript (Page:6, line 181).

**Author response image 3. sa2fig3:**